# Simulation and Application of Water Environment in Highly Urbanized Areas: A Case Study in Taihu Lake Basin

**Pengxuan Zhao [1,2], Chuanhai Wang [1,2,3], Jinning Wu [4], Gang Chen [1,2,3,*], Tianshu Zhang [1,2], Youlin Li [1,2] and Pingnan Zhang [1,2]**

1   The National Key Laboratory of Water Disaster Prevention, Hohai University, Nanjing 210098, China; pengxuanzhao@hhu.edu.cn (P.Z.); chwang@hhu.edu.cn (C.W.); 221301010115@hhu.edu.cn (T.Z.); 211601010033@hhu.edu.cn (Y.L.); pingnanzhang@hhu.edu.cn (P.Z.)
2   College of Hydrology and Water Resources, Hohai University, Nanjing 210098, China
3   Key Laboratory of Hydrologic-Cycle and Hydrodynamic-System of Ministry of Water Resources, Nanjing 210098, China
4   Changzhou Branch of Jiangsu Province Hydrology and Water Resources Survey Bureau, Changzhou 213022, China; wjnxx@163.com
*   Correspondence: gangchen@hhu.edu.cn; Tel.: +86-139-1302-9378

**Abstract:** In the wake of frequent and intensive human activities, highly urbanized areas consistently grapple with severe water environmental challenges. It becomes imperative to establish corresponding water environment models for simulating and forecasting regional water quality, addressing the associated environmental risks. The distributed framework water environment modeling system (DF-WEMS) incorporates fundamental principles, including the distributed concept and node concentration mass conservation. It adeptly merges point source and non-point source pollution load models with zero-dimensional, one-dimensional, and two-dimensional water quality models. This integration is specifically tailored for various Hydrological Feature Units (HFUs), encompassing lakes, reservoirs, floodplains, paddy fields, plain rivers, and hydraulic engineering structures. This holistic model enables the simulation and prediction of the water environment conditions within the watershed. In the Taihu Lake basin of China, a highly urbanized region featuring numerous rivers, lakes and gates, the DF-WEMS is meticulously constructed, calibrated, and validated based on 26 key water quality monitoring stations. The results indicate a strong alignment between the simulation of water quality indicators (WQIs) and real-world conditions, demonstrating the model's reliability. This model proves applicable to the simulation, prediction, planning, and management of the water environment within the highly urbanized watershed.

**Keywords:** distributed framework model; DF-WEMS; highly urbanized area; water environment; water quality model; Taihu Lake basin





## 1. Introduction

Cities, often situated along rivers, are closely tied to water, serving as landscape, economic, and cultural elements. In the rapid urbanization process in China, the expansion of urban scale and intensive production activities exert significant pressure on watershed water environments [1]. Water pollution emerges as a key constraint to the high-quality development of cities. Therefore, the establishment of a robust watershed water environmental model for simulating and predicting watershed water quality conditions holds significant importance in the realm of water environment protection and governance, particularly in highly urbanized areas [2].

From 1925 to the present, the theoretical development of water environmental models has undergone three distinct stages. The initial phase, spanning from 1925 to 1965, marked the inception of the Streeter–Phelps model [3]. During this period, the majority of water quality model programs underwent improvements based on the S-P model. These

programs focused on elucidating pollutant hydrodynamic transport processes in rivers and the intricate interplay among diverse water quality components. Eminent scientists, including Thomas, Dobbins, Camp, and O'Connor, progressively refined the S-P water quality model and developed corresponding models. In this stage, simulation indicators expanded from a singular parameter, Dissolved Oxygen (DO), to include other factors such as nitrogen–phosphorus cycling systems and planktonic plant and animal systems. Additionally, models evolved from simple one-dimensional structures to more intricate two-dimensional frameworks. The second stage, spanning from 1965 to 1995, witnessed the development of three-dimensional water quality models tailored for simulating the transformation and transport of pollutants in large water bodies. These models comprehensively considered interactions with sediments. The third stage, post-1995, saw the gradual establishment of pollution load models to simulate non-point source pollution, aiding governmental efforts in pollution control. Innovative methods, such as genetic algorithms, neural networks, and support vector machines, were incorporated into these models [4–7].

From 1925 to the present, water environmental model software systems have evolved from the Streeter–Phelps model, which initially could only simulate BOD and DO, to today's comprehensive model systems. Around 1970, the U.S. Environmental Protection Agency and the Geological Survey sequentially developed sophisticated water quality model software, including QUAL-I and WASP (Version 6.0). Similarly, European countries independently developed effective water quality models, such as the MIKE model by the Danish Hydraulic Institute, the DELFT-3D model by the Netherlands Delft Hydraulic Institute, and the ISIS model by the British Wallingford Software Company [8–13]. Recent years have witnessed continuous improvements in traditional water environment models, with simulation indicators becoming increasingly diverse. Many models now integrate exceptional hydrodynamic and water quality models, alongside outstanding result display tools. For example, the BASINS (Better Assessment Science Integrating Point and Nonpoint Sources) model system developed by the U.S. Environmental Protection Agency (USEPA) and Geological Survey (USGS) integrates the Environmental Fluid Dynamics Code (EFDC), Hydrological Simulation Program-FORTRAN (HSPF), TOXI-ROUTE river model, and QUAL2E water quality model [14–19]. This system comprehensively analyzes non-point source pollution loads, including nutrients, bacterial substances, and sediments in watersheds. The MIKE SHE model system developed by the Danish Hydraulic Institute, based on GIS, offers a comprehensive analysis of watershed water quantity and quality [20–24].

This paper introduces a distributed framework water environment model system (DF-WEMS) based on the distributed concept, tailored for simulating the entire watershed water environment in highly urbanized areas. Built upon a self-developed GIS system, DF-WEMS merges water quantity models, waste load models, and water quality models. Serving as a hydrodynamic–hydraulic–water-quality-merged model, DF-WEMS provides a comprehensive assessment of watershed water environment conditions. Additionally, the research independently develops a software system that facilitates rapid water environment modeling of watersheds. To verify the model's reliability, the paper selects the Taihu Basin, characterized by dense urban construction, a complex river network, and severe water environment problems, as the research area. The calibration and validation results demonstrate that DF-WEMS reliably and feasibly simulates water environment conditions in highly urbanized areas.

## 2. Study Area

The Taihu Lake Basin, situated in the southern part of the Yangtze River Delta in China, is surrounded by the Yangtze River to the north, the East China Sea to the east, and the Qiantang River to the south. Bounded by Tianmu Mountain and Maoshan Mountain in the west, the terrain generally slopes from west to east. Notably, it stands as a paradigm of high industrialization and urbanization in China, undergoing swift economic development. Despite occupying a mere 0.4% of China's territorial expanse and accommodating a mere

3% of China's population, this basin significantly contributes, constituting 10% of the national Gross Domestic Product (GDP) [25].

The topography of the watershed exhibits limited undulations, with a surface elevation differential of 1652 m and a watershed area spanning 36,895 km$^2$. The mean water depth within Taihu Lake is recorded at 1.9 m, encompassing a water surface area spanning 2338 km$^2$. The overall hydrological regime follows a west-to-east trajectory, with the majority of watercourses exhibiting flow velocities ranging from 0.1 m/s to 0.3 m/s. Land use within the basin is predominantly characterized by cropland, forest land, built-up areas, and water bodies. The basin is predominantly characterized by alluvial plains, comprising 66% of the total area, while aquatic surfaces account for 16%, and hilly to mountainous terrains cover 18% of the land. The hydroclimatic conditions within the basin exhibit characteristics of a subtropical monsoon climate. The multi-year mean temperature ranges from 15 °C to 17 °C. The multi-year average precipitation measures 1181 mm, with approximately 60% of the annual rainfall concentrated during the period from May to September, indicative of a distinct wet season. Prevalent sources of contamination include non-point source pollutants from agricultural activities and urban domestic effluents. The basin spans the provinces of Jiangsu, Zhejiang, Anhui, and Shanghai, making it one of the most densely populated and economically vibrant regions in China [26–30].

The basin boasts a well-developed water system, featuring an intricate network of rivers and lakes. The research area includes 547 main sewage outlets, comprising 255 industrial wastewater outlets, 106 domestic sewage outlets, and 186 mixed wastewater outlets. Annually, approximately 3.10 billion m$^3$ of wastewater flows into the river, contributing around 112,700 t/a of COD, 6100 t/a of NH$_3$-N, 11,000 t/a of TP, and 35,300 t/a of TN to the river. The distribution of sewage outlets is illustrated in Figure 1. The discharge of pollutants in the Taihu Lake basin significantly surpasses the water body's assimilative capacity, resulting in serious river and lake pollution and an overall pessimistic water quality status. Constructing a water quantity and quality model for the Taihu Lake basin to simulate the water quality conditions in the river and lake serves as crucial technical and decision-making support for comprehensive basin management. Figure 1 provides an overview of the study area's basic information.

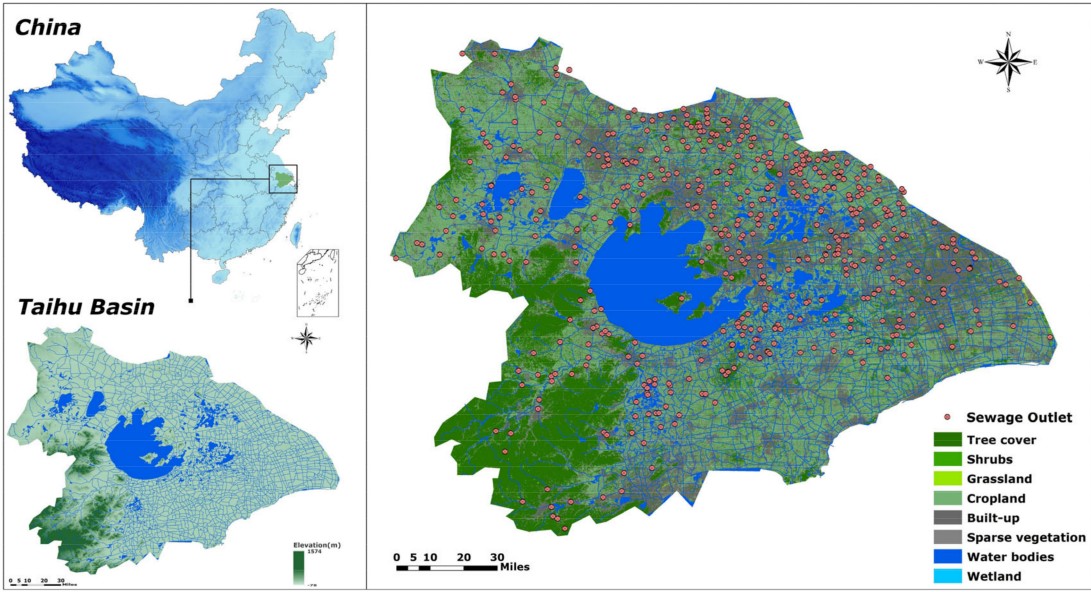

**Figure 1.** Geographical location, elevation, and distribution of sewage outlets and land use of the study area.

## 3. Materials and Methods

### 3.1. Waste Load Model

The computation of point source and non-point source pollution in the waste load model is bifurcated into two key components: the pollutant generation module and the pollutant treatment module (see Figure 2). The pollutant generation module comprises 8 distinct types of Pollutant Generation Units (PGUs) and operates with three calculation modes. Concurrently, the pollutant treatment module encompasses four types of pollutant treatment units (PTUs), each endowed with varying treatment efficiencies for diverse pollutants. Throughout the model calculation process, the pollutant generation module employs the relevant calculation mode to quantify the volume of pollution, considering the pollution generation characteristics of different PGUs. In alignment with the treatment efficiencies of various PTUs, the pollutant treatment module computes the quantity of pollutants entering the river network based on the pollution generated by different PGUs.

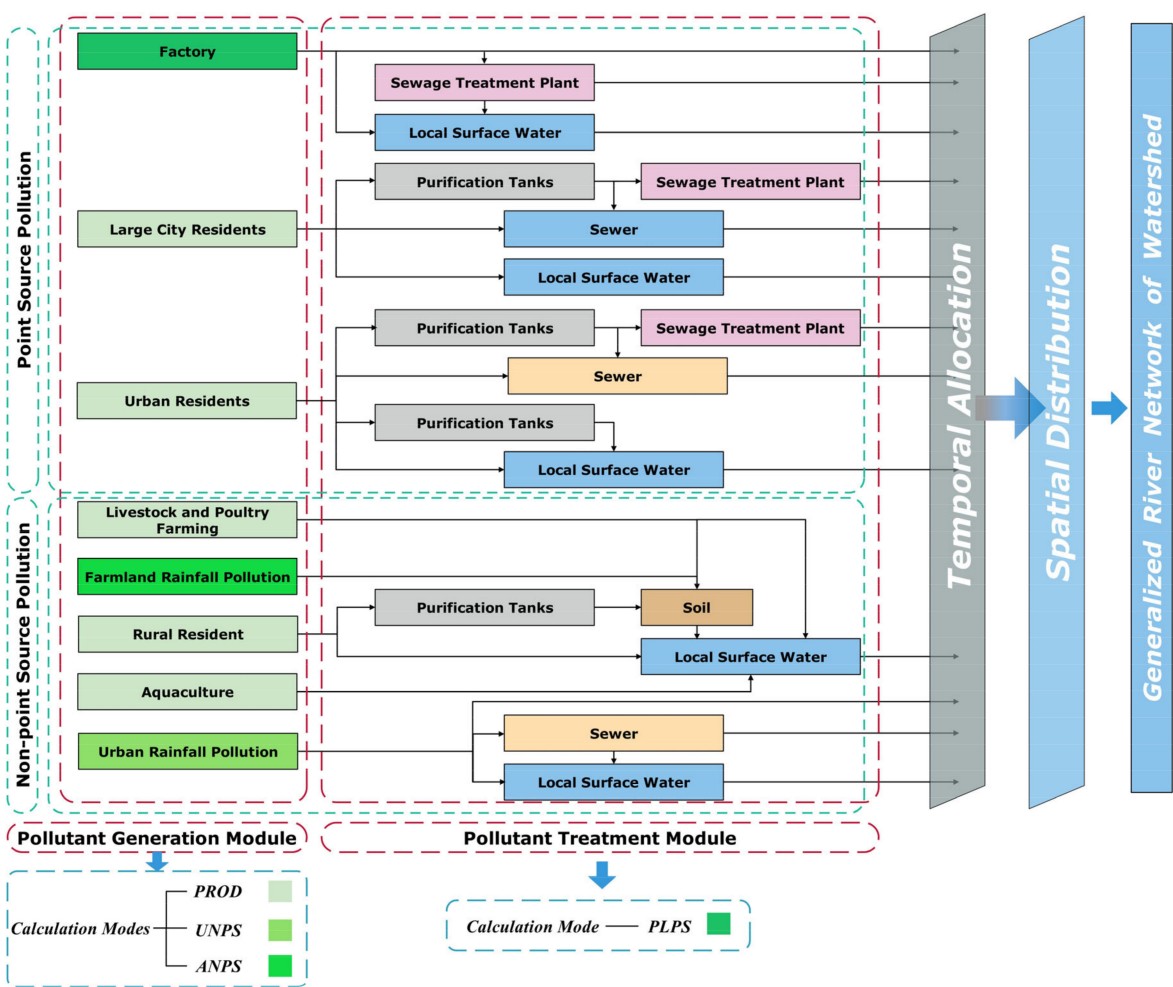

**Figure 2.** The pathways and processing procedures of pollutants in a watershed.

The waste load model bears resemblance to the water cycle model in the plain Hydrological Feature Unit (HFU) [31]. Pollutant Generation Units (PGUs) within this model encompass various sources, such as factories, large city residents, urban residents, livestock and poultry farming, farmland rainfall pollution, rural residents, aquaculture, and urban rainfall pollution (Figure 2). Pollutant Treatment Units (PTUs) consist of purification tanks, local surface water, sewage treatment plants, soil, and sewers. The pollutants generated by PGUs undergo treatment processes (e.g., degradation, sedimentation) through diverse pathways and PTUs, ultimately entering the river network. To calculate pollutant genera-

tion and the quantity of pollutants entering the river network, four calculation modes are established: PROD, (urban non-point source) UNPS, (agricultural non-point source) ANPS, and PLPS.

### 3.1.1. PROD Calculation Mode

The PROD calculation mode is applied to compute non-rainfall-related pollutant generation for four Pollutant Generation Units (PGUs): large city residents, urban residents, rural residents, livestock and poultry farming, and aquaculture. In the PROD model, Equation (1) is utilized to determine the quantity of pollution generation.

$$W_{\beta i}^{j} = N_i \cdot R_i^{j} \qquad (1)$$

where $W_{\beta i}^{j}$ is the production of the $j$th pollutant in the $i$th PGU (kg/a); $N_i$ is the number of the $i$th PGU; $R_i^{j}$ is the pollution load of the $j$th pollutant in the $i$th PGU (kg/a). The specific meaning of the variables in Equation (1) varies for different PGUs. For PGU of the large city residents, $N_i$ is the population of large cities and $R_i^{j}$ is the pollution equivalent of the population in large cities ($k_1$). For PGU of the urban residents, $N_i$ is the population of cities and $R_i^{j}$ is the pollution equivalent of the population in cities ($k_2$). For PGU of the rural residents, $N_i$ is the rural population and $R_i^{j}$ is the pollution equivalent of the rural population ($k_3$). For PGU of livestock and poultry farming, it divides livestock and poultry into four categories—cattle, pig, sheep and poultry—and $N_i$ is the number of livestock and poultry and $R_i^{j}$ is the pollution equivalent of livestock and poultry ($k_4$). For PGU of the aquaculture, $N_i$ is the number of the aquaculture and $R_i^{j}$ is the pollution equivalent of the aquaculture ($k_5$). For the practical calculation, socio-economic data are used for $N_i$ and the range of values for $R_i^{j}$ is shown in Table 1.

**Table 1.** Range of pollutant equivalence $k$ (kg/a) for different water quality indicators in each PGU.

| PGUs | COD | BOD$_5$ | TN | TP | NH$_3$-N |
|---|---|---|---|---|---|
| Large City Residents | 27.6–37.2 | 13–17.8 | 8.1–10.0 | 0.62–0.8 | 4.0–5.2 |
| Urban Residents | 18.7–28.0 | 9.2–15.3 | 7.5–10.0 | 0.4–0.6 | 3.1–5.2 |
| Rural Residents | 17.3–26.2 | 9.2–15.3 | 7.4–9.8 | 0.4–0.6 | 3.1–5.2 |
| Cattle | 223.4–337.4 | 149.0–247.0 | 51.8–69.3 | 7.2–11.0 | 19.6–32.7 |
| Pig | 23.9–36.2 | 20.0–33.1 | 3.7–4.9 | 1.7–2.7 | 1.7–2.87 |
| Poultry | 1.0–1.5 | 0.9–1.5 | 0.21–0.28 | 0.14–0.22 | 0.09–0.17 |
| Sheep | 4.0–5.98 | 2.1–3.4 | 3.5–4.7 | 1.0–1.6 | 0.4–0.7 |
| Aquaculture | 670.5–1012.8 | 117.0–193.7 | 85.6–114.5 | 7.9–12.1 | 14.0–23.4 |

### 3.1.2. UNPS Calculation Mode

The UNPS calculation mode is applied to compute pollution loads for both large cities and towns. In large cities, the model employs the pollutant accumulation-runoff scouring model to calculate pollution loads, whereas the average concentration method is utilized for estimating pollution loads in towns.

(1)    Calculation of Pollution Loads in Large Cities

The amount of pollutants carried by runoff is influenced by factors such as rainfall, runoff, and pollutant accumulation. The concept of daily critical precipitation is introduced to represent the threshold at which 90% of accumulated pollutants are scoured from the surface, occurring when the daily rainfall matches the daily critical precipitation. Equations (2)–(5) are employed to calculate the pollutant generation based on various land uses, facilitating the determination of the overall pollution load.

$$M = \begin{cases} P, & R_c = 0 \\ 0, & R_c > 0 \end{cases} \tag{2}$$

$$P = \sum_{i=1}^{n} P_i = \sum_{i=1}^{n} X_i A_i \tag{3}$$

$$X_i = \alpha_i F_i \, \gamma_i R_{cl} / 0.9 \tag{4}$$

$$\gamma_i = \begin{cases} N_i/20, & N_i < 20\,\text{h} \\ 1, & N_i \geq 20\,\text{h} \end{cases} \tag{5}$$

where $M$ is the daily accumulation of pollutants (kg); $R_c$ is daily urban rainfall (mm/d); $P$ is the total rate of pollutant accumulation (kg/d); $P_i$ is the rate of pollutant accumulation for the $i$th land use (kg/d); $X_i$ is the rate of accumulation of pollutants per unit area for the $i$th land use (kg/$\left(\text{km}^2{\cdot}\text{d}\right)$); $A_i$ is the total area of the $i$th land use (km$^2$); $n$ is the number of land use categories; $\alpha_i$ is the concentration parameter of urban pollutants (mg/L); $F_i$ is the parameter of population density; $\gamma_i$ is the parameter of ground sweeping frequency; $R_{cl}$ is the daily critical precipitation (mm/d); $N_i$ is the road sweeping time interval (d). Since the sweeping frequency in large cities is generally 1 time/day, the accumulation of urban surface pollutants does not exceed one day's accumulation. Table 2 displays the value of pollutant concentration parameters $\alpha_i$ obtained from experiments in Suzhou, Shanghai, Nanjing, Chongqing. The values of the population density parameter $F_i$ are displayed in Table 3 [32].

**Table 2.** The values of $\alpha_i$ of different pollutants in different urban land use types.

| Urban Land Use Types | COD (mg/L) | BOD$_5$ (mg/L) | TP (mg/L) | TN (mg/L) | NH$_3$-N (mg/L) |
|---|---|---|---|---|---|
| Living District | 14.0 | 3.5 | 0.15 | 0.58 | 0.174 |
| Commercial District | 56.4 | 14.1 | 0.33 | 1.31 | 0.393 |
| Industrial District | 21.2 | 5.3 | 0.31 | 1.22 | 0.366 |
| Other | 2.0 | 0.5 | 0.04 | 0.27 | 0.081 |

**Table 3.** The values of population density parameter $F_i$.

| Urban Land Use Types | Living District | Commercial District | Industrial District | Other |
|---|---|---|---|---|
| $F$ | $0.142 + 0.111D_P{}^{0.54}$, and $D_P$ is the population density (1/km$^2$) | 1.0 | 1.0 | 0.142 |

The quantity of pollutants scoured by runoff is dependent on factors such as rainfall intensity, duration, and sweeping frequency. Equations (6) and (7) apply the concept of first-order kinetics to calculate the amount of pollutants scoured by rainfall and runoff in urban areas.

$$\frac{dP}{dt} = -kR_s P \tag{6}$$

$$M_t = P\left(1 - e^{-kR_s t}\right) \tag{7}$$

where $P$ is the cumulative rate of pollutants for different land use types (kg/d); $k$ is the scouring coefficient (1/mm), generally ranging from 0.14 to 0.19; $R_s$ is the effective rainfall intensity (mm/h); $M_t$ is the scouring amount of surface pollutants when the rainfall duration is $t$ (kg).

(2)    Estimation of Pollution Load in Towns

In towns and nearby rural areas, nitrogen and phosphorus are significant regional nonpoint source pollutants. Therefore, when estimating pollution loads, the focus is solely on estimating nitrogen and phosphorus pollution loads. The estimation of pollution loads

in towns predominantly employs the method of average concentration. The concentrations of the Total Nitrogen (TN), Total Phosphorus (TP) and Ammonia Nitrogen (NH$_3$-N) in towns reach 7.26 ± 4.43 mg/L, 2.21 ± 0.90 mg/L and 1.16 ± 0.68 mg/L, respectively. The concentrations of TN and TP in the commercial and residential areas of the town are relatively low, yet notably higher than those observed in the surrounding surface water. Furthermore, these concentrations significantly surpass the V water quality standard for surface water.

### 3.1.3. ANPS Calculation Mode

The ANPS calculation mode is utilized to quantify pollution loads stemming from agricultural fields. This study employs distinct methodologies to calculate pollution generation from agricultural fields, distinguishing between two types of underlying surfaces: dryland and paddy fields.

(1)  Calculation of Pollution Load in Dryland

The correlation between pollutant loss per unit area of NH$_3$-N, TN and TP and effective rainfall depth is established (Equation (8)) by studying the loss rule of agricultural non-point source pollution in a typical small watershed (YiXing-MeiLin sub-watershed, in the hilly area around Taihu Lake).

$$q = \begin{cases} bR_d, & R_d < R_l \\ a\ln R_d + b, & R_d \geq R_l \end{cases} \tag{8}$$

Since the concentrations of Biochemical Oxygen Demand (BOD$_5$) and Chemical Oxygen Demand (COD) in surface runoff do not vary much with the rainfall duration, Equation (9) establishes a linear relationship between the amount of pollutant lost per unit area and the effective rainfall depth.

$$q = b \times Rd \tag{9}$$

where $q$ is the amount of pollution produced per unit area (kg/km$^2$); $R_d$ is the average daily effective rainfall depth (mm/d); $R_l$ is the critical value of effective rainfall depth (mm/d); $a$, $b$ are empirical parameters, which are determined by the experiment. Based on the results of Equations (8) and (9), the daily pollutant production for drylands in each calculation unit can be calculated using Equation (10).

$$WD_i = q_i \times A_{di} \tag{10}$$

where $q_i$ is the pollution generation per unit area corresponding to the daily effective rainfall depth $Rd_i$ in the $i$th calculation unit.

(2)  Calculation of Pollution Load in Paddy Field

The water generation process in paddy fields differs from that in drylands. Pollutants in paddy fields are discharged into local surface water only when the water depth of the paddy field exceeds the waterlogging tolerance of rice or during water abandonment. Additionally, nutrient exchange occurs at the water–soil interface due to the differing distribution of nitrogen (N) and phosphorus (P) elements at the interface. The N and P content in the soil spreads to the water body, leading to an increase in the N and P content in the water. After a period of diffusion, a dynamic equilibrium of adsorption and desorption is reached.

Based on the pollution generation mechanism of paddy fields, it is assumed that the paddy field has a certain initial water depth and initial pollutant concentration at the water surface. When daily precipitation is sufficiently mixed with the water surface, the water depth and pollutant concentration at the water surface of the paddy field will change. According to the laws of adsorption and desorption at the water–soil interface and the

assumption of sufficient mixing of pollutants, Equations (11) and (12) are established to calculate the concentration of pollutants in the surface water of paddy fields.

$$C_a^1 = \begin{cases} 0, & H_1 \leq H_e \\ \frac{H_0 C_a^0 + H_r C_r + H_i C_i}{H_1 + R_i} + \frac{C_{max} - C_a^0}{T}, & H_1 > 0 \end{cases} \tag{11}$$

$$WM_i = \begin{cases} 0, & R_i \leq 0 \\ 0.01 C_a \times R_i \times A_{mi}, & R_i > 0 \end{cases} \tag{12}$$

where $C_a^0$ is the concentration of pollutants in the surface water of the paddy field at the previous moment (mg/L); $C_a^1$ is the concentration of pollutants in the surface water of the paddy field at the latter moment (mg/L); $H_0$ is the water depth of the paddy field at the previous moment (mm); $H_1$ is the water depth of the paddy field at the latter moment (mm); $H_r$ is the rainfall during this period (mm); $C_r$ is the concentration of pollutants in rainwater (mg/L); $R_i$ is the effective rainfall depth of the paddy field (mm); $H_i$ is the irrigation water quantity in this period (mm); $C_i$ is the concentration of pollutants in irrigation water (mg/L); $C_{max}$ is the upper limit of pollutant concentration in the surface water the of paddy field (mg/L); $T$ is the release period of pollutants in the surface water of the paddy field (d); $C_a$ is the concentration of pollutants in the runoff generated by paddy field (mg/L); $WM_i$ is the daily amount of pollutants produced by paddy field (kg); $A_{mi}$ is the area of the paddy field in the calculation cell (km$^2$). $H_0$, $H_1$ and $H_r$ can be provided by the hydrological model and the value of $C_{max}$ can be referred to the experimental data of paddy fields in the southern part of Jiangsu Province, China. Table 4 displays the range of $C_r$ values.

**Table 4.** Reference values for $C_r$.

| WQI | Permanganate Index (mg/L) | BOD$_5$ (mg/L) | TN (mg/L) | TP (mg/L) | NH$_3$-N (mg/L) |
|---|---|---|---|---|---|
| Reference values for $C_r$ | 1.96–3.53 | 0.5–0.9 | 0.84 | 0.02 | 0.84 |

3.1.4. PLPS Calculation Mode

The PLPS calculation mode calculates the amount of pollutants entering the river network based on the amount of pollution load generated by the PGUs, the proportion of each pollution pathway, and the treatment efficiency of the PTUs, as shown in Equation (13).

$$W_e = W_i \times p_i \times (1 - f_i) \tag{13}$$

where $W_e$ is the amount of pollutants entering the river network (kg/d); $W_i$ is the amount of pollutant generation (kg/d); $p_i$ the proportion of different pollution pathways; $f_i$ is the treatment efficiency of the PTU. Table 5 displays the results of the generalization and estimation of the pollution pathways in the river network and their proportions $p_i$ based on the field research. The values of the treatment efficiency $f_i$ of different PTUs are displayed in Table 6. The treatment efficiency of different sewage treatment plants is determined by field research.

Within the hilly sub-watershed HFUs, a simple and efficient way to model non-point source pollution loads is employed, which is the statistical correlation model between rainfall-runoff and pollution loads, to calculate the pollutant output from the outlet cross-section of sub-watershed (Equation (14)).

$$E_a = aq^b \tag{14}$$

where $E_a$ is pollution loads per unit area (g/s·km$^2$); $q$ is the hydromodulus (m$^3$/s·km$^2$); $a$ and $b$ are parameters that can be calibrated using measured data in a particular watershed.

**Table 5.** The values of the generalization and estimation of the pollution pathways into the river network and their proportions $p_i$.

| PGUs | The Pollution Pathways | $p_i$ |
|---|---|---|
| Large City Residents | Large city residents to purification tanks to sewage treatment plants | 0.0–0.76 |
| | Large city residents to purification tanks to sewers | 0.14–0.9 |
| | Large city residents to sewers | 0.05 |
| | Large city residents to local surface water | 0.05 |
| Urban Residents | Urban residents to purification tanks to sewage treatment plants | 0.0–0.75 |
| | Urban residents to purification tanks to sewers | 0.05 |
| | Urban residents to sewers | 0.05 |
| | Urban residents to purification tanks to local surface water | 0.1–0.8 |
| | Urban residents to local surface water | 0.02–0.1 |
| Urban Rainfall Pollution | Urban rainfall pollution to river networks | 0.1 |
| | Urban rainfall pollution to sewer | 0.3 |
| | Urban rainfall pollution to sewer to local surface water | 0.5 |
| | Urban rainfall pollution to local surface water | 0.1 |
| Livestock and Poultry Farming | Livestock and poultry farming to soil to local surface water | 0.9 |
| | Livestock and poultry farming to local surface water | 0.1 |
| Farmland Rainfall Pollution | Farmland rainfall pollution to soil to local surface water | 1 |
| Rural Residents | Rural residents to purification tanks to soil to local surface water | 0.6 |
| | Rural residents to local surface water | 0.4 |
| Aquaculture | Aquaculture to local surface water | 1 |

**Table 6.** The values of $f_i$ of different PTUs and different pollutants.

| PTUs | COD (%) | BOD$_5$ (%) | TN (%) | TP (%) | NH$_3$-N (%) |
|---|---|---|---|---|---|
| Purification Tanks | 21–34 | 22–35 | 4–8 | 5–9 | −11−−20 |
| Sewer | 3–8 | 3.5–8 | 3–7 | 5–9 | 3.5–8 |
| Local Surface Water | 22–34 | 23–35 | 38–43 | 25–32 | 32–45 |
| Soil | 80–91 | 83–91 | 82–89 | 95–97 | 80–92 |

*3.2. Water Quality Model*

The plain river network area is typically distinguished by a multitude of rivers, lakes, and hydraulic engineering structures. The water flow direction in this region is often uncertain, contributing to the complexity of pollutant transport. Establishing a water quality model becomes crucial for analyzing pollutant transport in rivers, lakes, and hydraulic engineering structures. Such a model offers valuable technical support for the rational allocation of water resources in the plain river network area and facilitates the calculation of water environment capacity.

3.2.1. Pollutant Convective Transport Model of Plain River HFUs

The pollutant convective transport model for plain river Hydrological Feature Units (HFUs) is categorized into one-dimensional and two-dimensional water quality models within river networks. This paper focuses solely on elucidating the pertinent theories of the one-dimensional water quality model within river networks.

Equations (15) and (16) describe how the concentration of pollutants changes over time and space for a one-dimensional water quality model of river networks.

$$\frac{\partial(AC)}{\partial t} + \frac{\partial(UAC)}{\partial x} = \frac{\partial}{\partial x}\left(AE_x\frac{\partial C}{\partial x}\right) + \frac{AS}{86400} + S_w \tag{15}$$

$$E_x = \alpha_e C_0 \theta^2 q \tag{16}$$

where $A$ is the cross-sectional area of the water flow (m$^2$); $C$ is the concentration of WQI (mg/L); $t$ is the time (s) and $x$ is the spatial distance (m); $E_x$ is the longitudinal dispersion coefficient (m$^2$/s); $U$ is the average flow velocity of the cross-section (m/s); $S$ is the biochemical reaction term of a certain WQI (g/(m$^3$·d)); $S_w$ is an external source and sink term for a certain WQI (g/s); $\alpha_e$ is the coefficient, usually taken as 0.01; $C_0$ is the Chezy coefficient; $\theta$ is the ratio of the width and depth of the cross-section; $q$ is the average flow per unit width of the cross-section (m$^2$/s). Schematic diagram of one-dimensional channel controller is displayed in Figure 3.

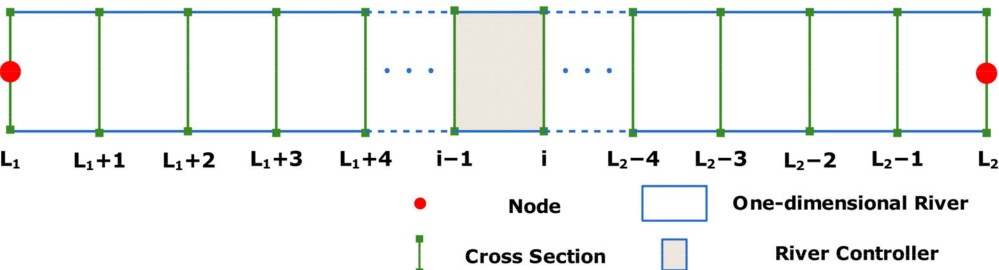

**Figure 3.** Schematic diagram of one-dimensional channel controller.

When solving the equations of the one-dimensional water quality model in practical applications, it becomes imperative to discretize the equations within the channel controller. The method of dividing channel controllers aligns with the water quantity model. Assuming a one-dimensional channel flows positively from $L_1$ to $L_2$, there are four primary modes of flow within a channel controller: water flowing into the channel controller from the positive direction, water flowing into the channel controller from the negative direction, water flowing into the channel controller from both ends, and water flowing out of the channel controller from both ends. The discretization of equations within the controller is grounded in three assumptions: mass conservation is maintained within the controller, the concentration within the controller varies linearly along $x$, and there are no negative waves in the downstream section of the controller.

Taking the situation where water flows into the channel controller in a positive direction as an example, each section of a one-dimensional river is divided into two very short sections on the left and right. The left section is the outflow section of the previous river microsection, and the right section is the inflow section of the latter river microsection. In practice, they are the same section, but this treatment is made for the convenience of calculation. There are two concentrations $Cl_i$ and $Cr_i$ at the left and right of the section $i$. In the channel controller between section $i-1$ and section $i$; when the time is $t_0$, the pollution concentration in section $i-1$ and section $i$ is $Cr_{i-1}^0$ and $Cl_i^0$, and the discharge is $Q_{i-1}$ and $Q_i$, and the concentration of the previous controller flowing into this controller is $Cl_{i-1}^0$. When time is $t_0 + \Delta t$, the wave is not transmitted to section $i$, and the concentration of section $i-1$ is equal to $Cl_{i-1}^0$. The actual concentration variation along $x$ is shown by the solid red line in Figure 4. If we follow the assumption that the concentration in the channel controller varies linearly along $x$, in order to maintain the mass balance of the controller, the concentration of section $i$ must be less than $Cl_i^0$ at this time (the outflow section produces a "negative wave"), and even the unreasonable phenomenon that the concentration value is less than zero. The fundamental reason for this unreasonable phenomenon is that the assumption of linear variation in concentration along $x$ in the channel controller is not in line with the actual situation. In fact, it is difficult to determine the variation in concentration along the channel controller, and in the simulation, it can only be assumed that the concentration changes linearly along the river. In order to avoid the unreasonable phenomenon mentioned above, it is assumed that the concentration of section $i-1$ at $t_0 + \Delta t$ is $Cr_{i-1}^{\Delta t}$, which is called the calculated concentration of the cross-section. $Cr_{i-1}^{\Delta t}$ is not the concentration of the upstream inflow but a "hypothetical value" based on the three

assumptions. Equation (17) expresses the increment of material transported to the channel controller through section $i-1$ after $\Delta t$ time.

$$M_1 = \left(Cl^0_{i-1} - Cr^0_{i-1}\right) \cdot Q_{i-1} \cdot \Delta t \tag{17}$$

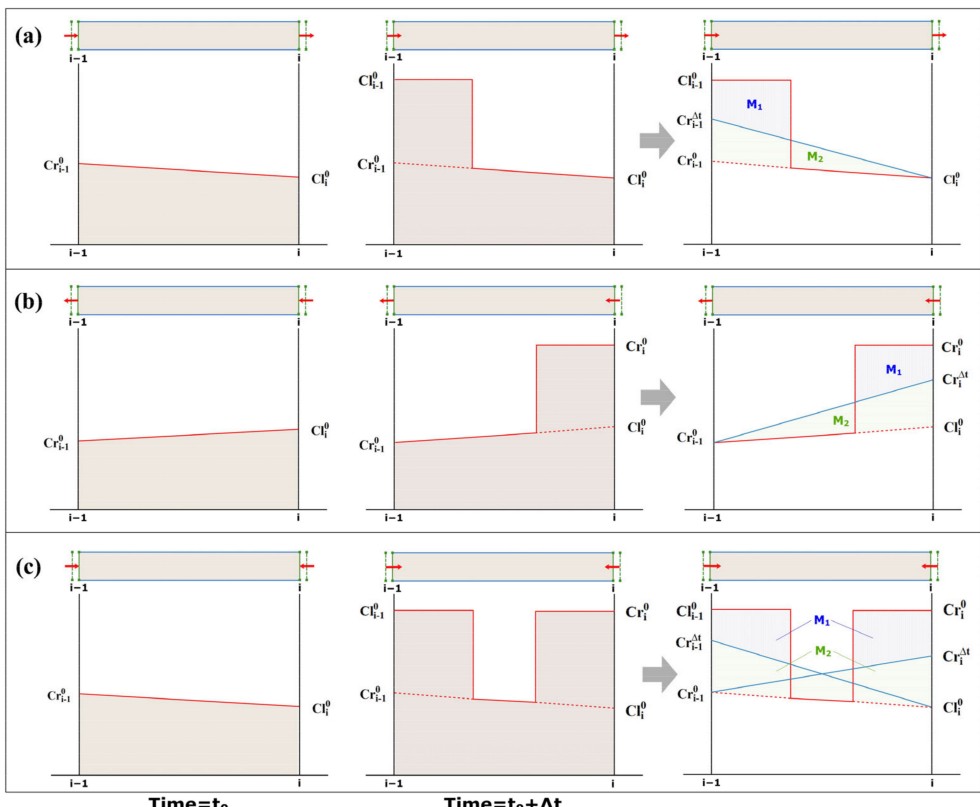

**Figure 4.** Schematic diagram of the discretization of the water quality model equations in the channel controller. (**a**) Water flowing into the channel controller from the positive direction; (**b**) water flowing into the channel controller from the negative direction; (**c**) water flowing into the channel controller from both ends.

To satisfy the three assumptions above, it is required that the amount of substance $M_2$ represented by the area of the triangle in Figure 4 be equal to $M_1$.

$$M_2 = 0.5(A_{i-1} + A_i) \cdot \Delta x \cdot \left(Cr^{\Delta t}_{i-1} - Cr^0_{i-1}\right) \tag{18}$$

Equation (19) for calculating the cross-sectional concentration $Cr^{\Delta t}_{i-1}$ can be obtained through Equations (17) and (18).

$$Cr^{\Delta t}_{i-1} = a_{i-1} + b_{i-1}Cl^0_{i-1} \tag{19}$$

In Equation (19), $a_{i-1} = (1 - \omega)Cr^0_{i-1}$, $b_{i-1} = \omega$, $\omega = \frac{2Q_{i-1}\Delta t}{(A_{i-1} + A_i)\Delta x}$. $\omega$ is an index reflecting the propagation speed of the wave; when $\omega < 1$, it indicates that the wave has not been transmitted to the downstream section, and the calculated concentration of section $i-1$ $Cr^{\Delta t}_{i-1}$ is between the initial concentration $Cr^0_{i-1}$ and the concentration of inflow into the channel controller $Cl^0_{i-1}$; when $\omega = 1$, the wave is just transmitted to the downstream section, and the calculated concentration of section $i-1$ $Cr^{\Delta t}_{i-1}$ equals to the concentration of inflow into the channel controller $Cl^0_{i-1}$; when $\omega > 1$, then the wave is just transmitted to the downstream section, and the calculated concentration of section $i-1$ $Cr^{\Delta t}_{i-1}$ is equal to the concentration $Cl^0_{i-1}$; when $\omega > 1$, take $\omega = 1$. In this case, the calculated concentration $Cr^{\Delta t}_{i-1}$

of section $i - 1$ is not the actual concentration of the section, but the assumption of linear change with the downstream section to calculate the amount of substance in the controller is correct, and it can be used to calculate the average concentration of the controller.

For Equations (15) and (16), the discrete form without considering the source-sink term, the biochemical reaction term, and the turbulent diffusion term are shown in Equation (20).

$$\chi Cr_{i-1}^{\Delta t} + \varphi Cl_i^{\Delta t} = W + Q_{i-1}Cl_{i-1}^0 \tag{20}$$

In Equation (20), $\chi = \frac{A_{i-1/2}^{\Delta t}\Delta x}{2\Delta t}$, $\varphi = \frac{A_{i-1/2}^{\Delta t}\Delta x}{2\Delta t} + Q_i$, $W = \frac{A_{i-1/2}^0\left(Cr_{i-1}^0 + Cl_i^0\right)\Delta x}{2\Delta t}$, $A_{i-1/2}^{\Delta t} = \frac{A_{i-1}^{\Delta t} + A_i^{\Delta t}}{2}$, $A_{i-1/2}^0 = \frac{A_{i-1}^0 + A_i^0}{2}$ and $A_{i-1/2}^{\Delta t}$ is the average discharge area of the river controller at time $t + \Delta t$, $A_{i-1/2}^0$ is the average discharge area of the controller at time $t$. Using Equations (18) and (20) can eliminate $Cr_{i-1}^{\Delta t}$ to obtain Equation (21) for calculating $Cl_i^{\Delta t}$.

$$Cl_i^{\Delta t} = \theta_i + \lambda_i Cl_{i-1}^0 \tag{21}$$

In Equation (21), $\theta_i = \frac{W - a_{i-1}\chi}{\varphi}$, $\lambda_i = \frac{Q_{i-1} - b_{i-1}\chi}{\varphi}$.

For the situation where water flows into the channel controller from the negative direction, similar processing can be performed to obtain Equation (22):

$$\begin{cases} Cl_i^{\Delta t} = a_i^* + b_i^* Cr_i^0 \\ Cr_{i-1}^{\Delta t} = \theta_{i-1}^* + \lambda_{i-1}^* Cr_i^0 \end{cases} \tag{22}$$

For the situation where water flows into the channel controllers from the two ends of the controller, Equation (23) can be derived.

$$\begin{cases} Cr_{i-1}^{\Delta t} = a_{i-1} + b_{i-1}Cl_i^0 \\ Cl_i^{\Delta t} = a_i^* + b_i^* Cr_i^0 \end{cases} \tag{23}$$

For the situation where water flows out of the channel controller from the two ends of the controller, the channel controller has no inflow flux from the upper and lower cross-sections during the calculation interval, and the concentration of pollutants in the controller is constant in the absence of the source-sink term, so that the concentration of pollution at the end of the time period depends on the amount of the substance at the beginning of the time period in the controller and the source-sink that has been added to and subtracted from the controller during the calculation interval, and the concentration in the controller can be described by the average concentration $\overline{C}$ as shown in Equation (24).

$$\begin{cases} Cr_{i-1}^{\Delta t} = \overline{C} \\ Cl_i^{\Delta t} = \overline{C} \end{cases} \tag{24}$$

In the river in Figure 4, there are $L_2 - L_1 + 1$ cross-sections; each cross-section has two unknown variables $Cl_i$ and $Cr_i$, and $2(L_2 - L_1)$ discrete equations can be established.

$$\begin{cases} Cr_i = \alpha_i + \beta_i Cl_{L_1} + \gamma_i Cr_{L_2} & (i = L_1, L_1 + 1, \ldots, L_2 - 1) \\ Cl_i = \xi_i + \eta_i Cl_{L_1} + \delta_i Cr_{L_2} & (i = L_1 + 1, L_1 + 1 \ldots, L_2) \end{cases} \tag{25}$$

where $\alpha_i$, $\beta_i$, $\gamma_i$, $\xi_i$, $\eta_i$, $\delta_i$ is the coefficient; $Cl_{L_1}$, $Cr_{L_2}$ is the concentration of the corresponding nodes. By solving Equation (25), the values of $Cl_i$ and $Cr_i$ for each cross-section of a one-dimensional river channel can be obtained.

### 3.2.2. Pollutant Convective Transport Model of Lakes and Reservoirs (Including Flood Plains and Paddy Fields) HFUs

Lakes and reservoirs, which include flood plains and paddy fields, play a crucial role in regulating and storing floodwaters, and they serve as significant sites for both pollutant

transport and degradation. Within this category of Hydrological Feature Units (HFUs), water quality models are categorized into zero-dimensional and two-dimensional models.

(1) Zero-Dimensional Water Quality Model

Zero-dimensional objects are generalized as nodes with regulation and storage effects on floods, and Equation (26) is used to describe it.

$$\frac{d(VC)}{dt} = \frac{VS}{86400} + S_w \tag{26}$$

where $C$ is the concentration of a certain WQI (mg/L); $V$ is the volume of water at the node of regulation and storage (m$^3$); $S$ is the biochemical reaction term for a certain WQI (g/(m$^3$·d)); $S_w$ is the source-sink term (g/s).

(2) Two-Dimensional Water Quality Model

Equation (27) describes the transport and transformation of pollutants in the two-dimensional object.

$$\frac{\partial(hC)}{\partial t} + \frac{\partial}{\partial x}\left(hUC - hE_x\frac{\partial C}{\partial x}\right) + \frac{\partial}{\partial y}\left(hVC - hE_y\frac{\partial C}{\partial y}\right) = \frac{hS}{86400} + S_w \tag{27}$$

For HFUs such as lakes, reservoirs, flood plains and paddy fields, the area is wide and extensive, with roughly equal scales of length and width, so the discretization of Equation (27) is performed without the use of coordinate transformations, and directly in the rectangular coordinate system, as opposed to the two-dimensional flow calculations in this type of HFUs [33], and the form of its cell is exhibited in Figure 5.

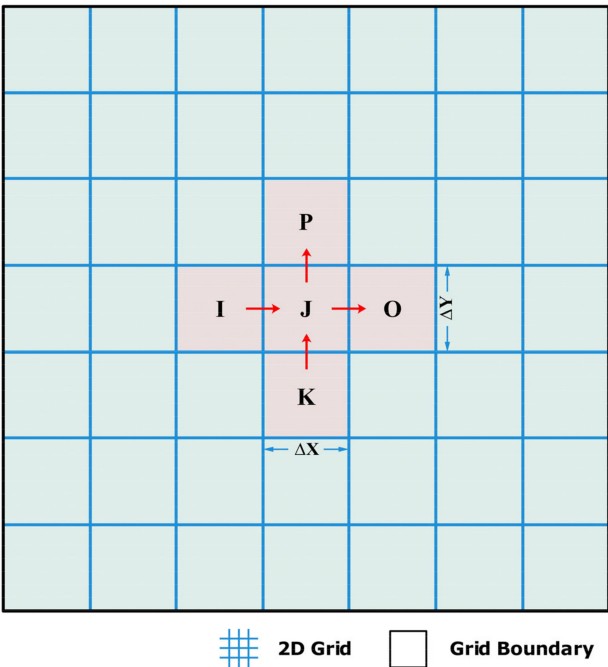

**Figure 5.** Schematic diagram of two-dimensional differential cell.

The finite volume method is used for discretization of cell $J$, and the pollutant flux input from cell $I$ to cell $J$ is $E_{I \rightarrow J}$:

$$E_{I \rightarrow J} = 0.5h_{IJ}U_{IJ}\left\{\left(1 + \delta_{(U_{IJ})}\right)C_I + \left(1 - \delta_{(U_{IJ})}\right)C_J\right\} - h_{IJ}E_x\frac{C_J - C_I}{\Delta X} \tag{28}$$

where $h_{IJ}$ is the water depth at the interface between cells $I$ and $J$; $U_{IJ}$ is the flow velocity at the interface between cells $I$ and $J$; $C_I$ and $C_J$ represents the concentrations of cells $I$ and

*J*, respectively; $\delta_{(U_{IJ})}$ is sign function. Similarly, the fluxes from unit *J* to unit *O*, from unit *K* to unit *J*, and from unit *J* to unit *P* are $E_{J\to O}$, $E_{K\to J}$, $E_{J\to P}$:

$$E_{J\to O} = 0.5 h_{JO} U_{JO} \left\{ \left(1 + \delta_{(U_{JO})}\right) C_J + \left(1 - \delta_{(U_{JO})}\right) C_O \right\} - h_{JO} E_x \frac{C_O - C_J}{\Delta X} \tag{29}$$

$$E_{K\to J} = 0.5 h_{KJ} U_{KJ} \left\{ \left(1 + \delta_{(U_{KJ})}\right) C_K + \left(1 - \delta_{(U_{KJ})}\right) C_J \right\} - h_{KJ} E_x \frac{C_J - C_K}{\Delta Y} \tag{30}$$

$$E_{J\to P} = 0.5 h_{JP} U_{JP} \left\{ \left(1 + \delta_{(U_{JP})}\right) C_J + \left(1 - \delta_{(U_{JP})}\right) C_P \right\} - h_{JP} E_x \frac{C_P - C_J}{\Delta X} \tag{31}$$

Using the finite volume method to discretize the two-dimensional water quality model equation, Equation (32) can be obtained.

$$\frac{h_J^{\Delta t} C_J^{\Delta t} - h_J^0 C_J^0}{\Delta t} + \frac{E_{J\to O} - E_{I\to J}}{\Delta X} + \frac{E_{J\to P} - E_{K\to J}}{\Delta Y} = \frac{h_J S}{86400} + S_w \tag{32}$$

Substituting the fluxes $E_{I\to J}$, $E_{J\to O}$, $E_{K\to J}$, $E_{J\to P}$ into Equation (33) yields a linear equation for the concentrations of the units *I*, *J*, *K*, *O*, *P*.

$$F_J\left(C_I, C_J, C_K, C_O, C_P\right) = 0 \tag{33}$$

Likewise, the comprehensive concentration equation can be formulated for the entire two-dimensional region. It is crucial to emphasize that, for boundary elements, the mass conservation equation must be integrated into the flux equation at the boundary interface. This integration establishes a connection between the nodes within the two-dimensional region and those outside of it.

3.2.3. Pollutant Convective Transport Model of Hydraulic Engineering Structures HFUs

Defining the flow of hydraulic engineering structures from node *I* to node *J*, Equation (34) can be used to calculate the exchange of pollutants between nodes *I* and *J*. The schematic diagram of the channel controller with hydraulic engineering structure is shown in Figure 6.

$$E_{I\to J} = \begin{cases} QC_J, & Q < 0 \\ QC_I, & Q \geq 0 \end{cases} \tag{34}$$

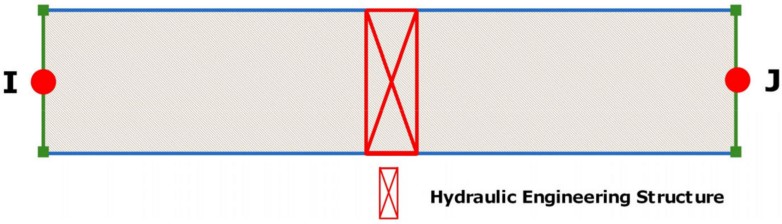

**Figure 6.** The channel controller with hydraulic engineering structure.

*3.3. The Coupling of Model*

3.3.1. Spatial Distribution of Pollution Loads

For point source pollution, its spatial distribution is discharged into the river network based on the principle of proximity. In other words, the pollutant is discharged into the nearest river.

The river system in the plain region exhibits a network characteristic, where the generalized river network and various boundary lines collectively form a closed polygonal area (Figure 7), known as the river network polygon. Due to the relatively small elevation difference in the plain area, employing a digital elevation model (DEM) for the spatial allocation of non-point source pollution within the river network polygons proves challenging. Consequently, based on the distributed concept and leveraging our self-developed Geographic

Information System (GIS), the calculation area is rasterized, and the calculation of pollutant generation within the raster is conducted, followed by the allocation of pollutants within the raster to the surrounding rivers using different weighting schemes. Taking a cell in Figure 7 as an example, the distances from the cell to the surrounding rivers are $d_1$, $d_2$, $d_3$, $d_4$, $d_5$. The length and corresponding cross-sectional area of the surrounding river are $L_1$, $L_2$, $L_3$, $L_4$, $L_5$ and $A_1$, $A_2$, $A_3$, $A_4$, $A_5$. Using Equation (35), we can calculate the allocation weight of non-point source pollution.

$$P_i^k = \frac{A_k/d_i^k}{\sum_{k=1}^m \left(A_k/d_i^k\right)} \tag{35}$$

where $P_i^k$ is the pollution allocation weight from the $i$th cell to the $k$th river (%); $d_i^k$ is the distance from the $i$th cell to the $k$th river (km); $A_k$ is the cross-sectional area of the $k$th river (m$^2$); $m$ is the number of rivers that form the river network polygon. This weighting calculation method considers both the impact of flow capacity and the structural characteristics of river network polygons on the spatial distribution of nonpoint source pollutants. After rasterizing, the land use of the underlying surface is categorized into four types: paddy fields, water surface, dry land and non-cultivated land, and urban areas. The area of each land use type in every raster is assigned to the surrounding rivers based on their respective weights. Subsequently, the total area of the four land use types within the river network polygons assigned to the rivers is compiled, ultimately deriving the pollution loads entering the rivers.

$$A_i^{jk} = A_i^j \times P_i^k \tag{36}$$

$$A^{jk} = \sum_{i=1}^n A_i^{jk} \tag{37}$$

$$W^k = \sum_{j=1}^4 \left(\varphi_j \times A^{jk}\right) \tag{38}$$

where $A_i^{jk}$ is the area allocated to the $k$th river by the $j$th type of land use in the $i$th raster (km$^2$); $A_i^j$ is the area of the $j$th type of land use in the $i$th raster (km$^2$); $A^{jk}$ is the area allocated to the $k$th river channel by the $j$th type of land use in the river network polygon (km$^2$); $W^k$ is the generation of pollution loads from the four land uses within the river network polygon into the $k$th river (kg/d); $\varphi_j$ is the pollutant transport flux of the $j$th type of land use (kg/(km$^2$·d)).

### 3.3.2. Temporal Allocation of Pollution Loads

The temporal allocation of point source pollution involves averaging over the year, meaning the annual value of a specific pollutant is averaged across the months. In contrast, non-point source pollution is transported through rainfall and runoff. The temporal distribution of non-point source pollutant inputs to the river is calculated utilizing the UNPS and ANPS calculation models.

### 3.3.3. Coupling of Water Quality Model

Throughout the watershed, the waste load model calculates the generation of point source and non-point source pollution, distributing the pollutant generation in both time and space to depict the process of pollutants entering the river. This information serves as a known input condition for the equations of the water quality model. Corresponding to the coupling of the water quantity model [33,34], the coupling of the water quality model encompasses the integration between the one-dimensional river and the zero-dimensional and two-dimensional lakes, reservoirs, flood plains, and paddy fields. The nodal concen-

tration stands as a pivotal element in the water quality model coupling. Equation (39) represents the mass conservation equation of the node.

$$\sum QC + \frac{VS}{86400} = \frac{VC - V^0C^0}{\Delta t} \tag{39}$$

When the node has no regulation and storage effect, $V = 0$.

Substituting Equations (25), (33) and (34) into Equations (26), (32) and (39), the linear equations about the concentration of nodes are constructed and solved by matrix identification method. After the node concentration is obtained, the concentration of cross-sections of the river network can be obtained.

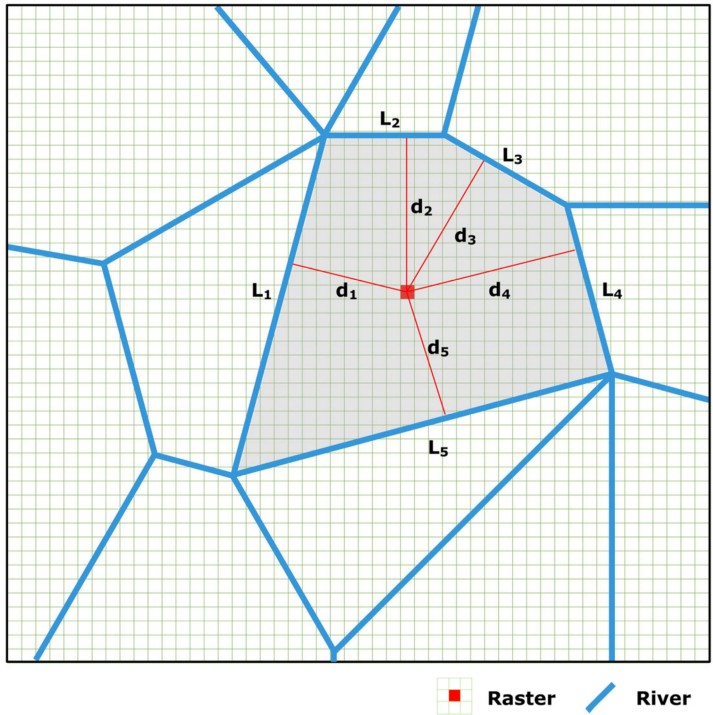

**Figure 7.** Schematic diagram of the spatial distribution of pollution load.

### 3.4. The Case Study

In the context of the Taihu Lake basin's water quantity model [16–19,35], A comprehensive water quantity and quality coupling model has been established to simulate the intricate water environment dynamics of rivers and lakes in the region. This model encapsulates HFUs for pollution generation and concentration across 26 plains and 10 hills, encompassing 129 zero-dimensional lakes designed for flood regulation and storage, 1481 one-dimensional rivers, and one two-dimensional lake. Additionally, 188 hydraulic engineering structures are considered in the model.

The model integrates point source pollution processes into the corresponding one-dimensional river calculation sections or two-dimensional lake calculation cells based on the geographical locations of point source pollution incidents (Figure 8a). For non-point source pollution areas, the model rasterizes these zones, automatically generating river network polygons to calculate pollution generation within the cells. Subsequently, the pollution produced within the cell is allocated to the rivers forming the river network polygon with varying weights (Figure 8b).

In total, the model incorporates 547 major point source pollutions, estimating generation coefficients, treatment efficiencies, and the proportion of each pollution pathway for different pollutants based on field research. The model defines 201 water quality boundary conditions, with monitoring stations providing monthly data for water quality indicators

(WQIs) such as Dissolved Oxygen (DO), Biochemical Oxygen Demand (BOD$_5$), Chemical Oxygen Demand (COD), Total Phosphorus (TP), Total Nitrogen (TN), and Ammonia Nitrogen (NH$_3$-N). Water temperature data utilize monthly values from 30 temperature monitoring stations in the Taihu Lake area, while wind speed and direction data are sourced from hourly meteorological data from the DongTingXiShan meteorological stations. Sunlight data are based on daily records from Shanghai, Hangzhou, and Nanjing.

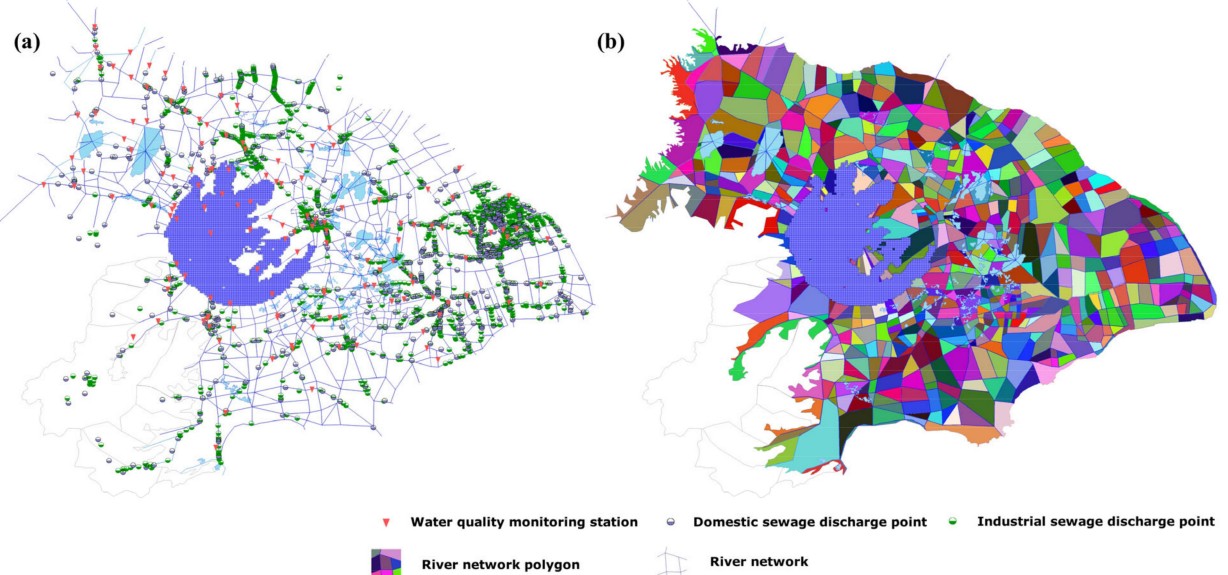

**Figure 8.** (**a**) Distribution of generalized river network, main point source pollution, and water quality monitoring stations; (**b**) model-generated river network polygons for the calculation of pollution concentration.

The spatial distribution of key stations for water quality boundaries, as well as monitoring stations for water temperature and meteorological data, is illustrated in Figure 9. The calculation period spans from 1 January 2012, to 31 December 2013, with 2013 data employed for model calibration and 2012 data for validation. Initial model conditions are established using measured water quality data from January, and specific initial conditions for the Taihu Lake area are determined according to measured data in different lake areas.

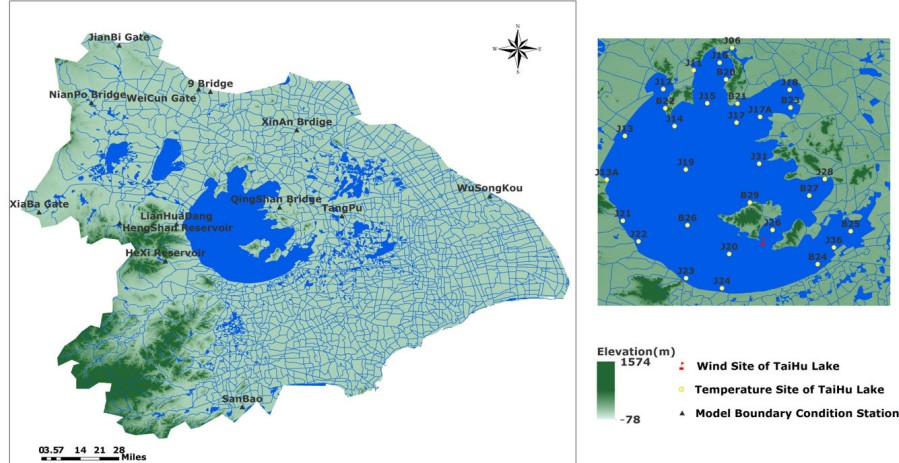

**Figure 9.** The distribution of water temperature stations, meteorological monitoring stations, and main reference stations for model boundary conditions.

## 4. Results

The DO, BOD$_5$, COD, NH$_3$- N, TP, and TN were gathered from 26 pivotal water quality monitoring stations, such as Tao Lake. These datasets were employed for model calibration, utilizing the absolute value of the relative error of the annual mean (AREAM) as the standard criterion. In the context of model validation, six WQIs are selected from a pool of 20 water quality monitoring stations, notably CaoQiao, situated in the key water function region. The validation process is conducted using AREAM as the benchmark. Furthermore, the simulation involved six WQIs from nine stations, including DaYi Bridge, to substantiate the model's efficacy in process simulation. These nine stations are situated at critical monitoring sections within the watershed's prioritized water quality monitoring zones. For example, the TaiPu Gate station, positioned on the main stem of the watershed, serves as a vital water quality monitoring station and plays a pivotal role in the supply of water to downstream areas, particularly Shanghai. The spatial arrangement of model calibration and validation stations is depicted in Figure 10. Overall, the selection of stations and WQIs for calibration and validation is methodologically sound and aligns effectively with the requisite criteria for model calibration and validation. Detailed AREAM values for WQIs at each station during the calibration and validation phases are tabulated in Tables 7 and 8.

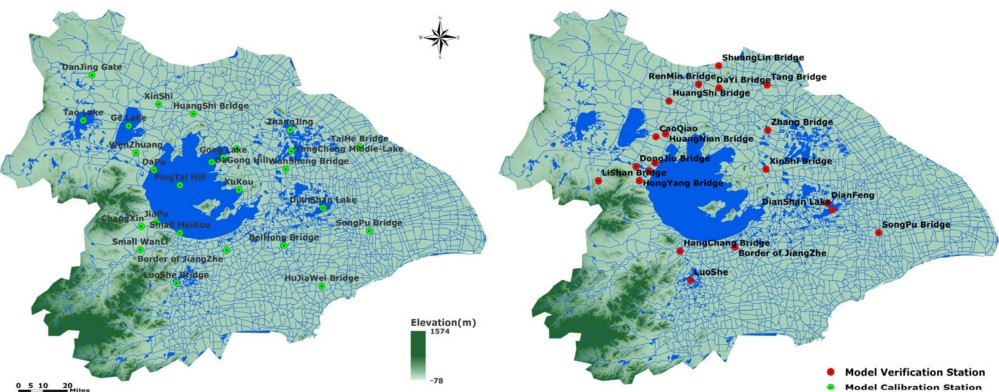

**Figure 10.** Spatial distribution of model calibration and validation stations.

From the calibration results (Tables 7 and 9), the average AREAM for DO across the 26 water quality monitoring stations is 13.31%. For BOD$_5$, the average AREAM is 14.59%, while for COD, it is 10.41%. NH$_3$-N exhibits an average AREAM of 14.85%, TP records 16.33%, and TN shows an average AREAM of 13.90%. The maximum AREAM value for each WQI is 46.97%, with 90% of AREAM for each WQI maintained within 30%.

Upon examining the verification results (Tables 8 and 9), the average AREAM for DO across the 20 water quality monitoring stations situated in key water functional areas is 17.84%. BOD$_5$ demonstrates an average AREAM of 10.13%, COD exhibits 12.97%, NH$_3$-N records 18.93%, TP shows 15.22%, and TN indicates an average AREAM of 15.43%. Notably, 85% of AREAM for DO falls within 30%, and 100% of AREAM for BOD$_5$, COD, and TN are within the 30% threshold. Moreover, 95% of AREAM for NH$_3$-N falls within 30%, and 90% of AREAM for TP is within 30%. Table 9 shows the maximum, minimum and median of AREAM (%) of different WQIs at model calibration and validation stations.

Figure 11 illustrates a boxplot depicting the AREAM in model calibration and validation. In conjunction with Table 9, it is discerned that, at the calibration stations, the median of AREAM for each WQI ranges from 10% to 15%, with the minimum AREAM falling below 1.5%. Conversely, at the validation stations, the median of AREAM for various WQIs ranges from 9.15% to a maximum of 21.67%. The minimum AREAM for each water quality indicator at validation stations is within 3%, while the maximum reaches around 60%. Notably, the model exhibits superior simulation efficacy for BOD$_5$ at validation stations, whereas the simulation for TP demonstrates a more moderate performance. In summary, errors at validation stations predominantly stay within 20%, and at calibration sites, errors

are generally constrained within 30%. The overall model simulation performance for diverse WQIs is deemed satisfactory.

**Table 7.** The absolute value of the relative error of the annual mean of water quality indicators of stations used for model calibration.

| Station | The Absolute Value of the Relative Error of the Annual Mean (%) | | | | | |
|---|---|---|---|---|---|---|
| | DO | BOD$_5$ | COD | NH$_3$-N | TP | TN |
| Tao Lake | 0.81 | 17.18 | 15.27 | 12.23 | 19 | 13.16 |
| HuJiaWei Bridge | 15.03 | 13.3 | 11.16 | 16.07 | 15.84 | 19.07 |
| WenZhuang | 7.27 | 24.97 | 10.04 | 19.9 | 9.75 | 4.52 |
| HuangShi Bridge | 22.83 | 2.33 | 0.34 | 5.85 | 10.07 | 6.96 |
| ZhangJing | 25.55 | 13.77 | 16.41 | 12.02 | 19.69 | 29.96 |
| WangTinLiJiao Gate | 10 | 7.86 | 14.83 | 2.15 | 11.81 | 0.85 |
| TaiHe Bridge | 11.26 | 3.87 | 6.8 | 28.94 | 7.74 | 0.69 |
| YangCheng Middle-Lake | 10.21 | 32.92 | 13.22 | 26.13 | 36.91 | 17.13 |
| DianShan Lake | 16.65 | 1.32 | 6.78 | 19.07 | 18.69 | 14.89 |
| SongPu Bridge | 2.04 | 0.52 | 3.6 | 2.35 | 6.46 | 2.99 |
| Border of JiangZhe | 18.92 | 5.52 | 1.05 | 20.49 | 39.73 | 6.11 |
| BeiHong Bridge | 36.66 | 37.91 | 27.94 | 19.68 | 46.97 | 14.48 |
| WanSheng Bridge | 30.56 | 4.59 | 3.27 | 30.29 | 10.5 | 0.34 |
| XinShi | 36.36 | 41.76 | 1.78 | 31.27 | 7.11 | 23.22 |
| LuoShe Bridge | 3.64 | 14.98 | 29.14 | 7.62 | 17.8 | 21.66 |
| ChangXin | 11.76 | 6.73 | 13.32 | 0.64 | 10.66 | 13.49 |
| Small WanLi | 18.15 | 8.1 | 3.76 | 26.33 | 9.03 | 20.45 |
| Gong Lake | 23.3 | 1.01 | 12.01 | 6 | 6.22 | 25.09 |
| DaPu | 14.18 | 5.77 | 2.28 | 15.31 | 3.47 | 9.22 |
| PingTai Hill | 2.02 | 12.09 | 16.63 | 3.58 | 14.74 | 23.76 |
| XuKou | 0.73 | 20.52 | 12.01 | 16.73 | 20.91 | 18.52 |
| Small MeiKou | 9.2 | 19.81 | 5.12 | 7.21 | 23.36 | 14.75 |
| DaGong Hill | 8.17 | 6.44 | 4.48 | 13.6 | 1.5 | 25.83 |
| Ge Lake | 5.38 | 41.83 | 12.2 | 3.23 | 14.68 | 9.35 |
| JiaPu | 5.24 | 6.35 | 12.6 | 27.66 | 27.39 | 10 |
| DanJing Gate | 0.21 | 27.9 | 14.74 | 11.86 | 14.54 | 14.84 |

In the validation period, the simulation process of six WQIs across nine stations, including CaoQiao, HuangNian Bridge, and TaiPu Gate, as illustrated in Figures 12–18, exhibits a consistent trend with the observed processes. In the context of process simulation for the nine strategically focused water quality monitoring stations within the watershed, computations are conducted to determine the average Relative Standard Deviation (RSD), the average Correlation Coefficient (R), and the average root mean square deviation (RMSD) between the measured and simulated values for various WQIs at these nine stations. A Taylor diagram (Figure 18) is generated for visual representation. Figure 18 reveals that the simulation performance excels particularly in the case of DO and BOD$_5$ at the highlighted stations, while the simulation efficacy for COD is moderately satisfactory. Moreover, the mean RSR (ratio of RMSE to the standard deviation of the observations) for various water quality indicators at the nine monitoring stations has been computed (Table 10). For conventional hydrological simulations, a model is generally considered acceptable when RSR falls within the range of 0 to 0.7 [36]. However, in the realm of integrated hydrological, hydraulic, and water quality simulations, particularly in expansive river networks, achieving uniformly low RSR values presents inherent difficulties. Consequently, this study adopts a more flexible criterion, deeming the overall model performance acceptable when RSR ranges from 0 to 1.0. Table 10 clarifies that the model established in the study area demonstrates RSR values below 1.0 for WQIs, excluding COD, at the specified key water quality monitoring stations. Notably, the RSR for NH$_3$ attains the lowest value at 0.63, while the RSR for COD slightly exceeds 1.0. Nonetheless, on the whole, the simulation outcomes

for all six water quality indicators are considered acceptable. The model demonstrates commendable performance in simulating water quality.

**Table 8.** The absolute value of the relative error of the annual mean of water quality indicators of stations used for model validation.

| Station | The Absolute Value of the Relative Error of the Annual Mean (%) | | | | | |
| --- | --- | --- | --- | --- | --- | --- |
| | DO | BOD$_5$ | COD | NH$_3$-N | TP | TN |
| CaoQiao | 26.04 | 5.24 | 19.49 | 23.34 | 15.22 | 2.79 |
| TaiPuGang Bridge | 5.37 | 5.37 | 20.25 | 29.60 | 7.97 | 23.08 |
| DaYi Bridge | 60.64 | 5.28 | 29.03 | 26.96 | 24.06 | 8.21 |
| Border of JiangZhe | 24.08 | 0.84 | 1.31 | 14.05 | 57.77 | 24.14 |
| DianFeng | 11.28 | 4.48 | 8.81 | 8.08 | 15.28 | 27.52 |
| DianShan Lake | 9.8 | 12.58 | 3.74 | 11.98 | 1.33 | 28.73 |
| DongJiu Bridge | 8.18 | 1.77 | 12.95 | 25.67 | 25.21 | 15.61 |
| HangChang Bridge | 16.57 | 16.33 | 7.38 | 4.50 | 26.29 | 24.80 |
| HongYang Bridge | 24.23 | 19.5 | 24.25 | 15.52 | 0.78 | 20.03 |
| HuangNian Bridge | 2.97 | 9.03 | 15.02 | 14.12 | 7.66 | 5.23 |
| HuangShi Bridge | 32.24 | 4.44 | 6.18 | 16.55 | 5.75 | 5.45 |
| LiShan Bridge | 7.72 | 7.97 | 23.97 | 20.82 | 18.68 | 21.58 |
| LuoShe | 18.75 | 18.68 | 26.71 | 30.60 | 38.00 | 1.67 |
| RenMin Bridge | 11.56 | 10.47 | 6.37 | 29.50 | 2.68 | 15.64 |
| SheDuGang Bridge | 19.17 | 19.28 | 0.46 | 27.28 | 25.94 | 13.00 |
| ShuangLin Bridge | 15.5 | 12.37 | 25.22 | 27.03 | 27.21 | 13.93 |
| SongPu Bridge | 8.14 | 5.75 | 2.11 | 0.64 | 3.86 | 22.63 |
| Tang Bridge | 15.31 | 15.31 | 1.87 | 27.14 | 11.49 | 14.11 |
| XinShi Bridge | 30.14 | 18.67 | 18.72 | 2.73 | 27.74 | 1.06 |
| Zhang Bridge | 9.27 | 9.26 | 2.07 | 22.52 | 1.56 | 19.32 |

**Table 9.** The statistical characteristics of AREAM (%) of different WQIs at model calibration and validation stations.

| Statistical Characteristics of AREAM(%) | | DO | BOD$_5$ | COD | NH$_3$-N | TP | TN |
| --- | --- | --- | --- | --- | --- | --- | --- |
| | Max | 36.66 | 41.83 | 29.14 | 31.27 | 46.97 | 29.96 |
| Calibration | Median | 10.74 | 10.1 | 11.59 | 14.46 | 14.61 | 14.62 |
| | Min | 0.21 | 0.52 | 0.34 | 0.64 | 1.50 | 0.34 |
| | Max | 60.64 | 19.50 | 29.03 | 30.6 | 57.77 | 28.73 |
| Validation | Median | 15.41 | 9.15 | 10.88 | 21.67 | 15.25 | 15.63 |
| | Min | 2.97 | 0.84 | 0.46 | 0.64 | 0.78 | 1.06 |

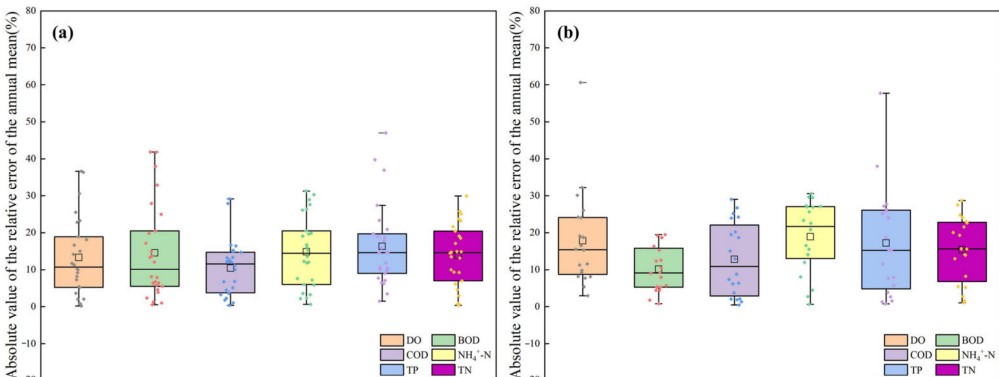

**Figure 11.** Box-plot of AREAM (**a**) Distribution of the calibration error (**b**) Distribution of the validation error.

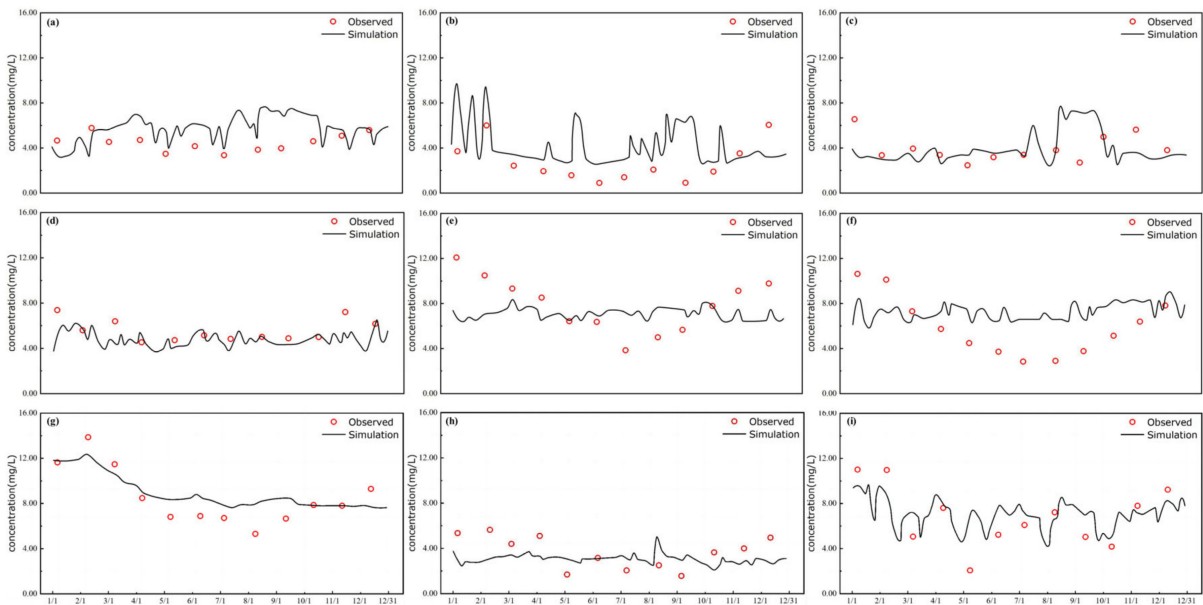

**Figure 12.** Comparison of simulated and measured values of DO at stations of (**a**) CaoQiao; (**b**) DaYi Bridge; (**c**) HuangNian Bridge; (**d**) HuangShi Bridge; (**e**) LiShan Bridge; (**f**) Border of JiangZhe; (**g**) TaiPu Gate; (**h**) Tang Bridge; (**i**) Zhang Bridge.

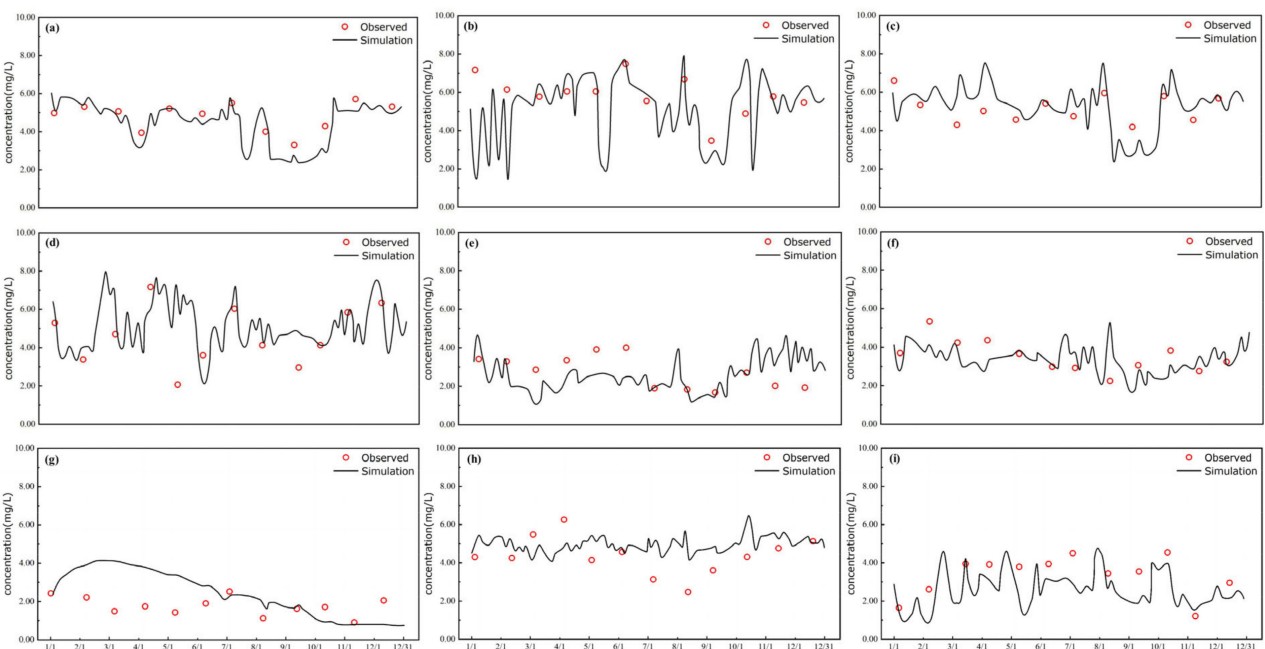

**Figure 13.** Comparison of simulated and measured values of $BOD_5$ at stations of (**a**) CaoQiao; (**b**) DaYi Bridge; (**c**) HuangNian Bridge; (**d**) HuangShi Bridge; (**e**) LiShan Bridge; (**f**) Border of JiangZhe; (**g**) TaiPu Gate; (**h**) Tang Bridge; (**i**) Zhang Bridge.

**Table 10.** The average RSR of 9 strategically focused water quality monitoring stations.

|  | **DO** | **BOD** | **COD** | **NH$_3$** | **TP** | **TN** |
|---|---|---|---|---|---|---|
| RSR | 0.86 | 0.95 | 1.14 | 0.63 | 0.91 | 0.65 |

Operating as an advanced water quantity and quality model tailored for plains, this model incorporates the intricacies of pollution generation and concentration processes across complex underlying surfaces, along with the dynamics of numerous lakes, rivers,

and hydraulic engineering structures. It is acknowledged that the simulation of WQIs is influenced not only by the inherent errors in the water quantity model but also by inevitable discrepancies in the water quality model itself. Nevertheless, the simulation error of the water environment model, as established in this study, falls within an acceptable range, with simulated values closely aligning with measured values. The overall trend is notably consistent, accurately reflecting the real-world scenario.

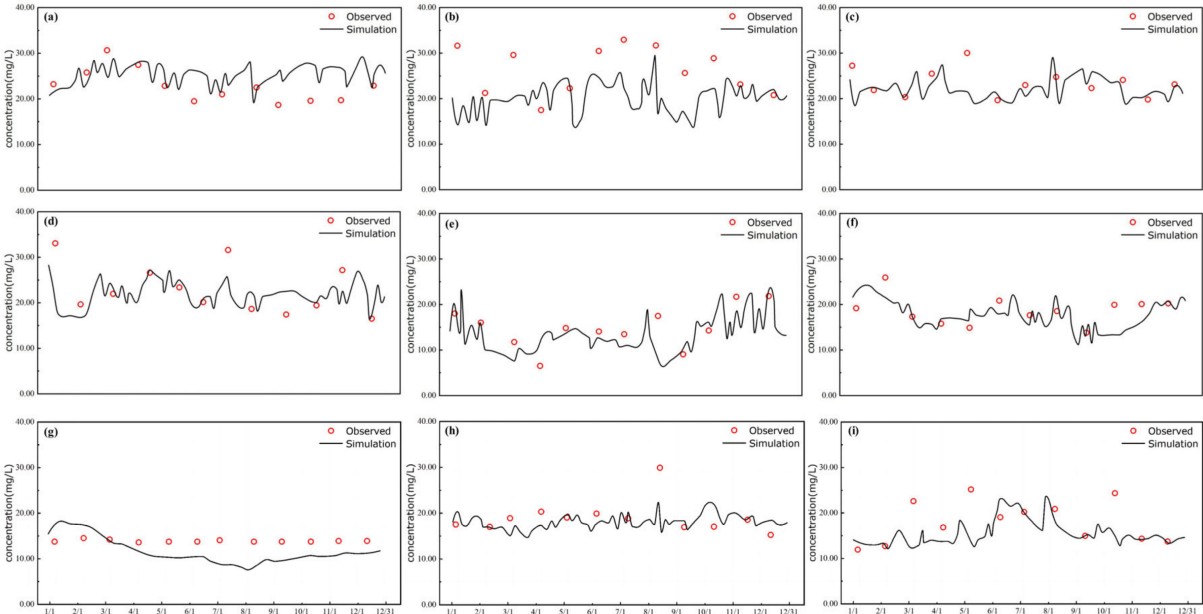

**Figure 14.** Comparison of simulated and measured values of COD at stations of (**a**) CaoQiao; (**b**) DaYi Bridge; (**c**) HuangNian Bridge; (**d**) HuangShi Bridge; (**e**) LiShan Bridge; (**f**) Border of JiangZhe; (**g**) TaiPu Gate; (**h**) Tang Bridge; (**i**) Zhang Bridge.

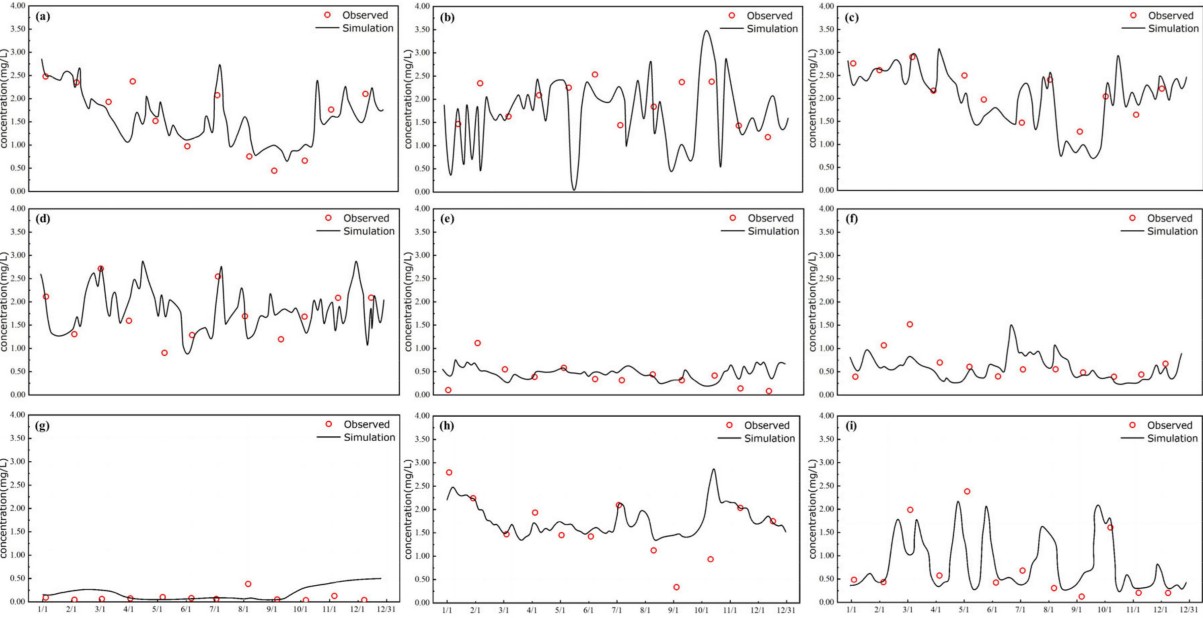

**Figure 15.** Comparison of simulated and measured values of NH$_3$-N at stations of (**a**) CaoQiao; (**b**) DaYi Bridge; (**c**) HuangNian Bridge; (**d**) HuangShi Bridge; (**e**) LiShan Bridge; (**f**) Border of JiangZhe; (**g**) TaiPu Gate; (**h**) Tang Bridge; (**i**) Zhang Bridge.

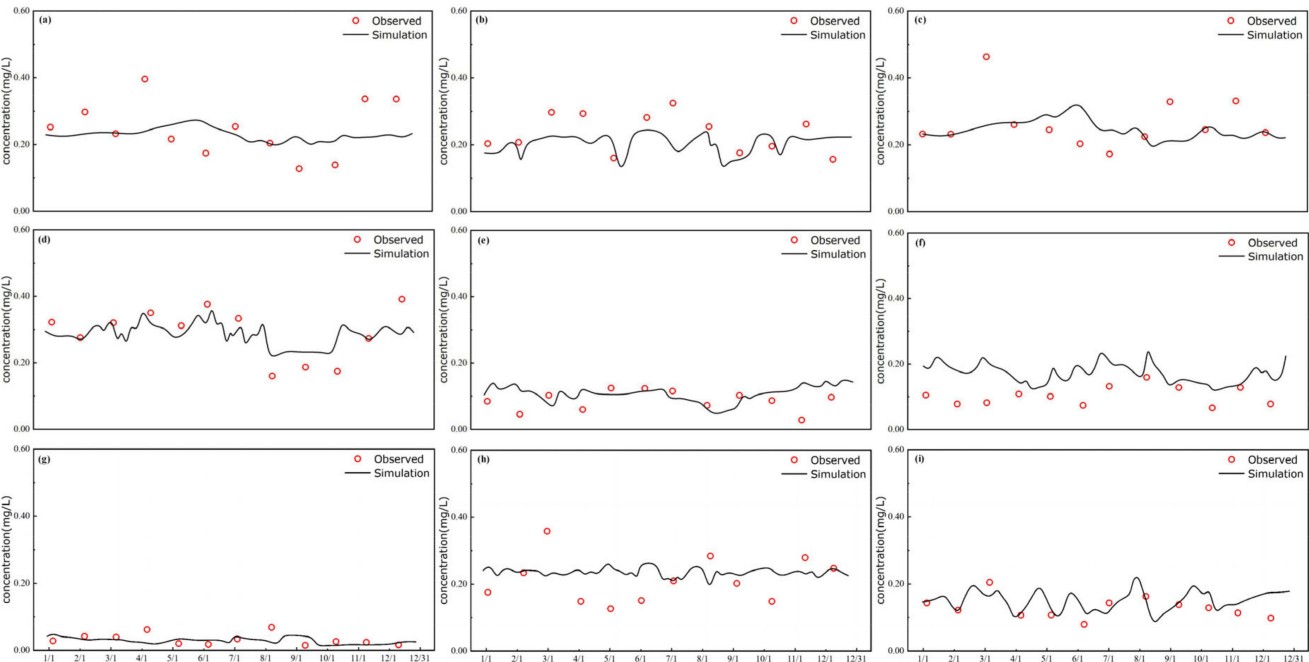

**Figure 16.** Comparison of simulated and measured values of TP at stations of (**a**) CaoQiao; (**b**) DaYi Bridge; (**c**) HuangNian Bridge; (**d**) HuangShi Bridge; (**e**) LiShan Bridge; (**f**) Border of JiangZhe; (**g**) TaiPu Gate; (**h**) Tang Bridge; (**i**) Zhang Bridge.

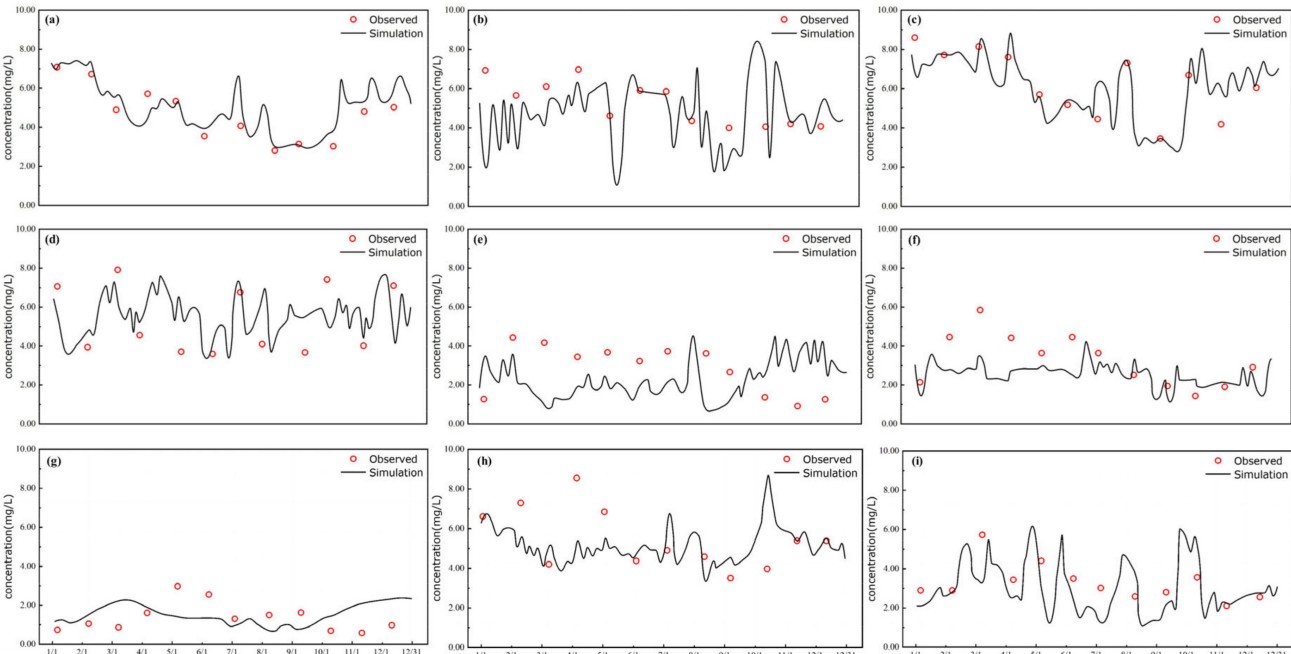

**Figure 17.** Comparison of simulated and measured values of TN at stations of (**a**) CaoQiao; (**b**) DaYi Bridge; (**c**) HuangNian Bridge; (**d**) HuangShi Bridge; (**e**) LiShan Bridge; (**f**) Border of JiangZhe; (**g**) TaiPu Gate; (**h**) Tang Bridge; (**i**) Zhang Bridge.

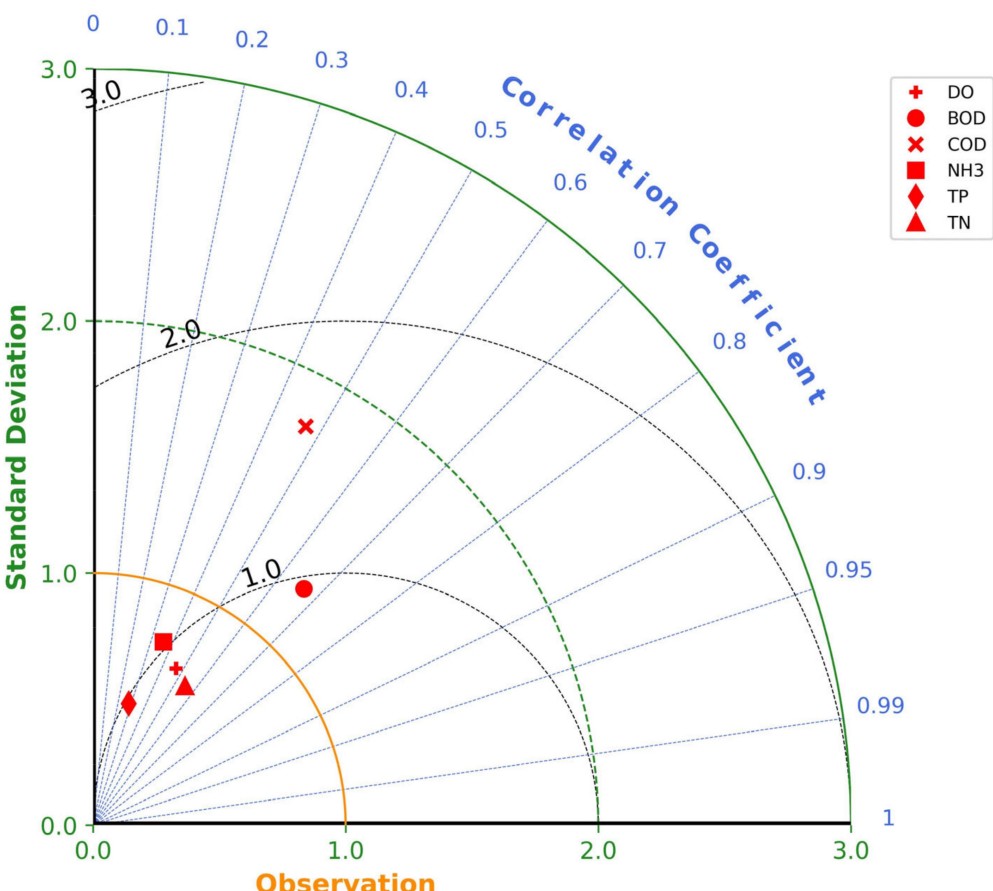

**Figure 18.** Taylor diagram of the different WQIs in 9 strategically focused water quality monitoring stations.

In conclusion, the model demonstrates a robust simulation effect on the water environment in highly urbanized regions. Its capacity to accurately capture the dynamics of water quality indicators underscores its utility as a valuable tool for comprehending and managing water quality in complex urban settings.

## 5. Discussion

In the present investigation, an exhaustive examination is undertaken concerning waste load models, hydrological models, hydraulic models, and water quality models within the intricate river network regions. Special emphasis is placed on addressing the challenges associated with their interdependencies. Leveraging our proprietary Geographic Information System (GIS), a software system tailored for the expeditious modeling of watershed water environments is developed. Through the simulation of the water environment within the study area, this endeavor yields highly encouraging outcomes. Notably, the model demonstrates a commendable performance, with the overall error in the calibration and validation of stations falling within acceptable limits. At the nine focal monitoring stations across the watershed, the errors associated with water quality indicators, specifically the RSR and RMSD, are deemed satisfactory. This model stands poised to provide pivotal decision support for the effective management and planning of water environments within the watershed.

The primary objective of this study is to address the intricate challenges associated with the simulation and forecasting of water environments in highly urbanized regions characterized by complex underlying surfaces and intensified anthropogenic activities. The model is deployed in the expansive and highly urbanized river network of the Taihu Basin, yielding globally acceptable simulation results. The water quality simulation outcomes at the key monitoring stations are particularly noteworthy. A novel aspect of this study

involved the introduction of Hydrological Feature Units (HFUs), whereby the watershed is systematically partitioned into distinct units, encompassing plain rivers, lakes, hydraulic engineering structures, and other pertinent entities. Employing a distributed approach, the watershed is discretized into a grid structure, facilitating pollution simulations within grid cells and pollutant transport simulations between adjacent cells. In summary, this study has successfully established a distributed integrated watershed water environment simulation model, adeptly addressing the intricate water quality simulation challenges inherent in highly urbanized regions. With respect to the simulated water quality outcomes in the study area, a comprehensive analysis reveals that the AREAM at both calibration and validation stations fell within acceptable bounds. The simulated processes of WQIs at the nine strategically positioned water quality monitoring stations demonstrate a high degree of concordance with the corresponding empirical measurements.

The traditional water quality model systems face several challenges. For instance, SWAT (Soil Water and Analysis Tools) struggles to effectively manage and modify large watersheds, and WASP (Water Quality Analysis Simulation Program) cannot address simulation issues related to pollutants with sinking or floating characteristics [37–40]. In contrast, the model established in this paper takes into account pollutant generation from complex underlying surfaces and the pollutant confluence in intricate river networks with numerous hydraulic engineering structures such as gates and pumps. This model successfully achieves a refined simulation of water quality tailored for large watersheds. Notwithstanding the attainment of a certain degree of success in this research endeavor, it is imperative to acknowledge the methodological challenges that have surfaced. The model's performance in simulating COD and TP is not entirely satisfactory, potentially attributed to the intricate interplay of microorganisms within river and lake ecosystems—an aspect not explicitly considered during the modeling process. Furthermore, for the myriad and intricately interlinked river–lake systems characteristic of highly urbanized areas, the multitude of influencing factors and the dynamic nature of underlying surface conditions pose formidable challenges to the accurate simulation of water quantity and quality across the expansive watershed.

Consequently, a strategic pivot towards focusing on key sentinel stations, pivotal to the overall watershed water environment, is deemed both pragmatic and achievable with the current model. Achieving a high level of precision in simulating WQIs at all watershed stations necessitates a more meticulous foundation of accurate data and a more extensive repository of monitoring data. Additionally, the waste load model of the hilly sub-watershed HFUs is somewhat empirical. Exploring new simulation theories is a research avenue that needs further development in the future. Simultaneously, there is a need to explore the migration and transformation patterns of pollutants within the soil of plain river network areas, as well as the application of artificial intelligence in water quality simulation, to achieve a more precise simulation of water environments [41,42].

## 6. Conclusions

The DF-WEMS as a component of the Distributed Framework for Basin Management Systems (DFBMS) is a water quantity and quality coupling model built upon the foundations of DF-HMS and DF-RMS. Specifically designed for highly urbanized regions, DF-WEMS integrates hydrological, hydrodynamic, and water environment models. It comprehensively addresses the processes of runoff generation and concentration, pollutant convection and diffusion, and incorporates various hydraulic engineering structures within the watershed. This approach allows for a practical representation of the interrelation of key elements in the watershed. The model incorporates zero-dimensional, one-dimensional, and two-dimensional water quality models tailored to different Hydrological Feature Units (HFUs), such as lakes and reservoirs, plain rivers, flood plains, paddy fields, and hydraulic engineering structures. The equations are jointly solved, with node concentration as the key variable, yielding concentrations for one-dimensional cross-sections and two-dimensional vertical lines.

To validate the model's rationality, a water environment model is established in the highly urbanized Taihu Lake basin, characterized by numerous rivers, lakes, and a complex underlying surface. Calibration involves selecting six water quality indicators (WQIs) from 26 key water quality monitoring stations, while validation uses six WQIs from 20 monitoring stations. The absolute value of the relative error of the annual mean for model calibration and verification falls within an acceptable range. Moreover, the simulation of six WQIs in nine stations situated at critical monitoring sections within the watershed's prioritized water quality monitoring zones, including CaoQiao, aligns well with measured processes, affirming the model's reliability for simulating water environments in highly urbanized areas.

Due to challenges associated with factors such as the difficulty in accurately acquiring pollution source data and the limited availability of water quality measurements, the simulation of water quality models has consistently proven to be demanding. Moreover, the Taihu Basin, characterized by complex and unpredictable water flow dynamics, lacks sufficient measured flow data for effective calibration of water quantity models, thereby amplifying the intricacies of refining the mechanistic aspects of water quality models. While the outcomes of model calibration and validation in this study generally exhibit good accuracy, offering a fundamental portrayal of the spatial distribution characteristics of water quality in 2013 and 2012, certain limitations persist. To address these, potential measures include establishing a data-sharing mechanism with governmental bodies to enhance and update pollution source-related information, along with intensifying water quality monitoring data across temporal and spatial scales within the watershed. Overall, the DF-WEMS stands as a valuable tool for watershed simulation and management of water environments.

**Author Contributions:** Conceptualization, P.Z. (Pingnan Zhang) Investigation and methodology, T.Z. and Y.L. Formal analysis, software, and visualization, P.Z. (Pengxuan Zhao) and J.W. Writing—original draft preparation, G.C. Writing—review and editing. Supervision, G.C. and C.W. All authors discussed the results and commented on the paper and figures. All authors have read and agreed to the published version of the manuscript.

**Funding:** This research was financially supported by the National Natural Science Foundation of China (U2040209, U2240209), the National Key Research and Development Program of China (2023YFC3208704), Hydraulic Science and Technology Program of Jiangsu Province (2022003), and the Cooperative Innovation Center for Water Safety & Hydro Science (B2106017).

**Data Availability Statement:** No new data were created or analyzed in this study. Data sharing is not applicable to this article.

**Conflicts of Interest:** The authors declare no conflict of interest.

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
