# Peer review of "Simulation and Application of Water Environment in Highly Urbanized Areas: A Case Study in Taihu Lake Basin"

_hydrology, doi:10.3390/hydrology11020020_

Round 1

Reviewer 1 Report

Comments and Suggestions for Authors

The study aims regional water quality assessment under environmental changes due to human activities. For this aim, The Distributed Framework Water Environment Modeling System (DF-WEMS) is used in the paper. Water quality data are gathered from 26 gauging stations in Taihu Lake Basin, China. Then, WQIs are calculated for the application. The subject is very important and the study is valuable in terms of monitoring environmental changes in river basins but the novelty of the study is emphasized insufficiently. Some suggestions and comments to the authors are presented below:

1. The flowchart of the suggested methodology should be given properly and by more branches and in detail in Figure 1. Thus, the readers can easily follow the application procedures.

2. Some legends on the figures & maps should be presented better and live colours. Most of words, numbers, etc. can’t be read on them.

3. Conclusions part can be improved in the paper. Here is presented in a general concept. Especially, the selection procedures of gauging stations from pool for the validation step should be discussed in detail here.

4. What is the novelty of the paper? WQI calculation is explained in the paper. Supported and related studies should be strongly presented in the paper by emphasizing the novelty of the paper.

5. “Discussions” part can be separated from “Results and Discussions” in the paper. Also, literature part is looking weak. Give main and last updated examples from literature about “water quality” as

10.1016/j.ecolind.2023.110522

10.5505/pajes.2013.46855

10.1016/j.ecolind.2023.110522

10.1016/j.wse.2020.06.006

6. How did the authors select six validation stations from the pool? Is there selection rule at this step?

7. As one important step of the study, the statistical characteristics of used data (e.g. water quality data) should be presented in detail. The statistical properties as skewness, coefficient of variation, confidence intervals, distribution characteristics, min, max and median, etc. of used data should be given in a table.

8. The performance metrics part is weak in the paper; only relative error is calculated. More metrics can be calculated to evaluate the application results NSE, RSR (Ratio of RMSE to the standard deviation of the observations) etc. …

Comments on the Quality of English Language

Check the tenses of the sentences. There are present and past tenses in a paragraph. See the paragraph in the Abstract.

There are some crucial errors. There are long sentences, see lines 18-23 …

Keywords should be ordered from A to Z. One more keyword as “DF-WEMS” can be added to keywords.

Use passive sentences. Check the sentences starting with “we”. See the line 90 …

The direction names should be started by capital letters such as South, Southern, North…

Reviewer 2 Report

Comments and Suggestions for Authors

The paper is relevant and well structured. I suggest to accept it while taking into account the following remarks:

Introduction

There is a lack of clear goal of the study.

Materials and methods

1. Line 95. There are five units analyzed, not 4.

2. Table 1: How do COD and BOD can be measured in kg/a? These are oxygen demand, not a pollutant load. Also, not clear why these parameters were chosen. For water quality, other parameters are more valuable (nitrites, nitrates, sulfates, chlorides, etc.). Please add a justification.

3. Tables 2 and 3: Are the data obtained by authors themselves? If so, please indicate this. If no, please add a reference.

4. Lines 218 and 220. “… based on the field research.” What kind of field research were conducted and where?

5. Sections 2.1 and 2.2 seem to be too big and overloaded with mathematics. Since these are not your research results but known models, I suggest to shorten these sections by leaving only the main equations you use in the study.

6. Also, please add what a software / environment was used for modelling.

7. Lines 477-479. The model includes a huge amount of pollution sources. How do features of each source are taken into account in the model? I would suggest to clarify this.

Results

1. Lines 509-512. Is it correct to calibrate the model by one pool of stations and validate the model by other pool? In my opinion, both calibration and validation should be done for one pool of stations but in different time.

2. Line 529. Absolute Relative Error of 30% looks quite large. Is it really acceptable?

3. In Materials section there were 3 modes of calculation (PROD, UNPS, and ANPS). Which mode was used in Results section?

4. I would suggest to add more analysis of results (by figures) – at least to compare different pollutants – for which parameter the modelling was more relevant?

Reviewer 3 Report

Comments and Suggestions for Authors

There are many comments which the authors must make to improve the paper: 

1. Introduction section is very short and  needs updating for the results reached in the field of research, gabs and novelty of the present work.

2. Study area description (location, topography, climate .......) section must be added before Materials and methods.

3. Figure 3 not clear.

4.   Discussion must be separated from results.

5. Conclusion section needs to be clear and recommendations  for the future development in the area must be added.

6. References are limited and old (You can cite this reference: DOI 10.1007/s12517-017-2960-x; ) and others' related works. 

Reviewer 4 Report

Comments and Suggestions for Authors

Dear Authors, hydrology-2768096

Thank you for opportunity to review this manuscript, ‘Simulation and application of water environment in highly urbanized areas: a case study in Taihu Lake Basin.’ The authors experiment highly urbanized areas consist ently grapple with severe water environmental challenges, in the wake of frequent and intensive human activities. 

According to the authors, their results indicate a strong alignment between the simulation of water quality indicators (WQIs) and real-world conditions, demonstrating the model's reliability. The model proves applicable to the simulation, prediction, planning, and management of the water environment within the highly urbanized watershed. 

In this process, the logical structure of was well designed and organized in a proper way. I have no qualm to suggest this manuscript for acceptance in the journal ‘Hydrology.’

However, the authors should divide into discussions after results section. The results and discussions should not be the identical. And then, they need to address some minor typos and resize the table size into consistent format.

Comments on the Quality of English Language

Dear Authors, hydrology-2768096

Thank you for opportunity to review this manuscript, ‘Simulation and application of water environment in highly urbanized areas: a case study in Taihu Lake Basin.’ The authors experiment highly urbanized areas consist ently grapple with severe water environmental challenges, in the wake of frequent and intensive human activities. 

According to the authors, their results indicate a strong alignment between the simulation of water quality indicators (WQIs) and real-world conditions, demonstrating the model's reliability. The model proves applicable to the simulation, prediction, planning, and management of the water environment within the highly urbanized watershed. 

In this process, the logical structure of was well designed and organized in a proper way. I have no qualm to suggest this manuscript for acceptance in the journal ‘Hydrology.’

However, the authors should divide into discussions after results section. The results and discussions should not be the identical. And then, they need to address some minor typos and resize the table size into consistent format.

Round 2

Reviewer 1 Report

Comments and Suggestions for Authors

I suggest accepting the manuscript. The authors carefully revised the paper by answering each comment from the first round.

Reviewer 3 Report

Comments and Suggestions for Authors

Accepted